# PeakDecoder enables machine learning-based metabolite annotation and accurate profiling in multidimensional mass spectrometry measurements

Aivett Bilbao [1,2,9] ✉, Nathalie Munoz [1,2,9], Joonhoon Kim[1,2,9], Daniel J. Orton [1], Yuqian Gao [1,2], Kunal Poorey[3], Kyle R. Pomraning [1,2], Karl Weitz[1], Meagan Burnet[1], Carrie D. Nicora [1], Rosemarie Wilton[2,4], Shuang Deng[1,2], Ziyu Dai[1,2], Ethan Oksen [5], Aaron Gee[6], Rick A. Fasani[6], Anya Tsalenko[6], Deepti Tanjore[2,5], James Gardner[2,5], Richard D. Smith [1], Joshua K. Michener [2,7], John M. Gladden[2,3], Erin S. Baker [8], Christopher J. Petzold [2,5], Young-Mo Kim [1,2], Alex Apfel[6], Jon K. Magnuson [1,2] & Kristin E. Burnum-Johnson [1,2] ✉

Multidimensional measurements using state-of-the-art separations and mass spectrometry provide advantages in untargeted metabolomics analyses for studying biological and environmental bio-chemical processes. However, the lack of rapid analytical methods and robust algorithms for these heterogeneous data has limited its application. Here, we develop and evaluate a sensitive and high-throughput analytical and computational workflow to enable accurate metabolite profiling. Our workflow combines liquid chromatography, ion mobility spectrometry and data-independent acquisition mass spectrometry with PeakDecoder, a machine learning-based algorithm that learns to distinguish true co-elution and co-mobility from raw data and calculates metabolite identification error rates. We apply PeakDecoder for metabolite profiling of various engineered strains of *Aspergillus pseudoterreus*, *Aspergillus niger*, *Pseudomonas putida* and *Rhodosporidium toruloides*. Results, validated manually and against selected reaction monitoring and gas-chromatography platforms, show that 2683 features could be confidently annotated and quantified across 116 microbial sample runs using a library built from 64 standards.

Metabolomics is the study of the small molecules produced by complex networks of cellular processes and biochemical reactions in living systems. Metabolites are the end point of the flow of information from DNA to the biological phenotype and represent chemical fingerprints directly reflecting the physiological conditions, intracellular regulation, and effects that environmental factors induce in biological cells or organisms. As such, metabolomics helps in a variety of applications, from understanding disease progression

[1]Pacific Northwest National Laboratory, Richland, WA, USA. [2]US Department of Energy, Agile BioFoundry, Emeryville, CA, USA. [3]Sandia National Laboratory, Livermore, CA, USA. [4]Argonne National Laboratory, Lemont, IL, USA. [5]Lawrence Berkeley National Laboratory, Berkeley, CA, USA. [6]Agilent Research Laboratories, Agilent Technologies, Santa Clara, CA, USA. [7]Oak Ridge National Laboratory, Oak Ridge, TN, USA. [8]Department of Chemistry, University of North Carolina, Chapel Hill, NC, USA. [9]These authors contributed equally: Aivett Bilbao, Nathalie Munoz, Joonhoon Kim. ✉e-mail: Aivett.Bilbao@pnnl.gov; Kristin.Burnum-Johnson@pnnl.gov

in clinical settings to estimating overproduction for metabolic engineering[1,2].

Advances in synthetic biology, genome editing, and DNA synthesis capabilities have propelled the ability to routinely design and generate thousands of novel strains for biomanufacturing research. The Agile BioFoundry (ABF) consortium of national laboratories utilizes state-of-the-art capabilities within the framework of the design, build, test, and learn (DBTL) cycle to develop engineered organisms[3]. Accurate analytical tools with fast turnaround time in Test are critical in developing microorganisms that can produce desired fuels and chemicals from renewable biological feedstocks.

The most popular and widely used analytical platform for the analysis of metabolic species in complex mixtures is mass spectrometry (MS) combined with liquid chromatography (LC) or gas chromatography (GC) separations[2,4,5]. However, hundreds to thousands of primary and secondary metabolites in nature display a high degree of structural diversity with many isomers and nominal mass isobars that co-elute and have similar fragmentation patterns, all of which constitute a significant analytical challenge in terms of detection and annotation. The incorporation of several orthogonal technologies in MS-based workflows can provide heterogeneous information to tackle these challenges. In fact, experimental measures such as retention time (RT) from chromatography, collision cross-section (CCS) from ion mobility spectrometry (IM), or stable isotope labeling, are necessary to complement MS/MS similarity and add confidence in overall compound identification workflows[6].

Besides increasing annotation confidence, multidimensional LC-IM-MS workflows collecting extensive fragmentation spectra with data-independent acquisition (DIA) methods are providing heterogeneous information which allows deeper understanding in metabolomics studies. IM is a gas phase separation technique increasingly used to distinguish structurally similar molecules, isomers, and molecular classes in biological and environmental samples[7]. Unlike LC that separates molecules based on hydrophobicity, IM separates gas-phase molecular ions based on their charge, size, and shape, which improves selectivity and coverage compared to routine LC-MS-based methods.

In DIA the mass spectrometer is operated to systematically collect multiplexed fragment-ion spectra (MS2) from all detectable precursors (MS1) within a wide *m/z* range and in a single chromatographic run, independently of their intensities[8]. Like initially found in proteomics[9], in metabolomics the MS2 spectrum quality of ions that get selected during standard data-dependent acquisition (DDA) is higher, but the overall MS2 coverage and quantitative precision using DIA is better[10]. While DIA provides increased reproducibility and quantitation performance, it requires more elaborated processing algorithms compared to DDA. Two main DIA processing strategies initially established for proteomics have been adapted to metabolomics in a handful of DIA metabolomics tools. The first strategy applies untargeted feature detection followed by deconvolution of fragment ion spectra (here referred to as UFD). A popular tool used for UFD in metabolomics is MS-DIAL[11], which groups precursors and their corresponding fragments based on the similarity of their elution profiles, generates pseudo-MS2 spectra and matches them against a reference MS2 library. Other reported tools applying UFD are MetaboDIA[12] and DaDIA[13]. The second DIA algorithmic strategy employs targeted data extraction (here referred to as TDX). TDX requires a library of target analytes with retention times, and precursors with corresponding fragment masses, which are utilized as coordinates to mine the DIA spectra and generate extracted ion chromatograms (XIC) for precursor and fragments per target analyte, as the so-called 'peak-group'. Multiple sub-scores are then calculated per peak-group to assess coelution and identification. Software employing TDX include Skyline[14], MetDIA[15], and DIAMetAlyzer[16]. Another tool demonstrated for DIA using a different approach is DecoID[17], where the MS2 deconvolution is achieved by mixing database spectra to match an experimentally acquired spectrum using least absolute shrinkage and selection operator (LASSO) regression.

While these tools exist for DIA metabolomics, new tools capable to fully exploit all dimensions with controlled error rates in multidimensional LC-IM-MS measurements with DIA spectra are needed. Skyline and MS-DIAL were adapted to support the additional IM dimension but they do not provide a false-discovery rate (FDR) control method. Unlike proteomics, the field of metabolomics still lacks a generally accepted, validated, and automated calculation of error rates for MS2 compound identification with FDR assessments[18]. Several methods have been proposed to generate decoys and estimate FDR in metabolomics. For imaging-MS, pySM[19] generates decoys by using implausible ion adducts. For DDA, Passatutto[20] uses re-rooted fragmentation trees, JUMPm[21] adds a small odd numbers of hydrogen atoms, and XY-Meta[22] combines original and randomly selected MS2 peaks. And recently reported for DIA, DIAMetAlyzer[16], provides an FDR estimation employing Passatutto[20] but it does not support the IM separation. These methods rely on annotated spectra or a sample-specific metabolite database for FDR estimation.

Here, we develop a sensitive and high-throughput analytical and computational workflow that combines LC-IM-MS multidimensional measurements with PeakDecoder, an algorithm that automatically calculates error rates for metabolite identification, independently of spectral annotations or libraries. PeakDecoder proposes an alternative method for decoy generation from raw DIA spectra, incorporating concepts from DIA and spectral library searching into a machine learning (ML) strategy that combines both UFD and TDX. To illustrate our metabolomics workflow and demonstrate its utility, we apply it to study microbial samples from various strains engineered under projects of the ABF consortium.

## Results
### Optimizing the LC-IM-MS analytical method
We defined a list of 64 metabolites of interest for the study of various strains of *Pseudomonas putida*, *Aspergillus pseudoterreus, Aspergillus niger,* and *Rhodosporidium toruloides*, all relevant microorganisms in the biotechnology field for production of value-added chemicals. The panel consisted of metabolites from central carbon metabolism including glycolysis, tricarboxylic acid cycle (TCA) cycle intermediates, amino acids, and 'coenzyme A' molecules (CoAs) that are routinely analyzed in ABF studies to obtain an overview of changes in the metabolism of cells. Additionally, metabolites specific to ABF host-bioproduct pairs, meaning compounds that are directly along the engineered pathways were also included.

The microorganisms in this study are promising industrial hosts and have a variety of application interests. *P. putida* is a Gram-negative, rod-shaped bacterium that is metabolically versatile, tolerant to toxins and solvents, with a high supply of reducing power, making it ideal for numerous biomanufacturing applications[23]. The eukaryotic microorganisms *A. pseudoterreus* and *A. niger* (filamentous fungi) were modified for production of 3-hydroxypropionic acid, a polymer precursor that can be dehydrated to produce acrylic acid and can be used directly within existing infrastructure[24]. Similarly, the *R. toruloides* (oleaginous yeast) strains were engineered for production of bisabolene, which is a precursor to a diesel alternative and is considered an ideal platform for bioconversion of lignocellulose into lipids and related chemicals[25].

To select an LC method, we implemented the Automated chromatographic Method Selection Software (AMSS), which utilizes chemical and physical properties of metabolites to predict the LC method that maximizes the number of metabolites detected (see "Methods"). The evaluation of the selected metabolites using AMSS predicted HILIC with negative ESI as the best method (Supplementary Fig. 1). The LC conditions were first implemented and optimized by selected reaction monitoring (SRM) analyses of a subset of the standards and

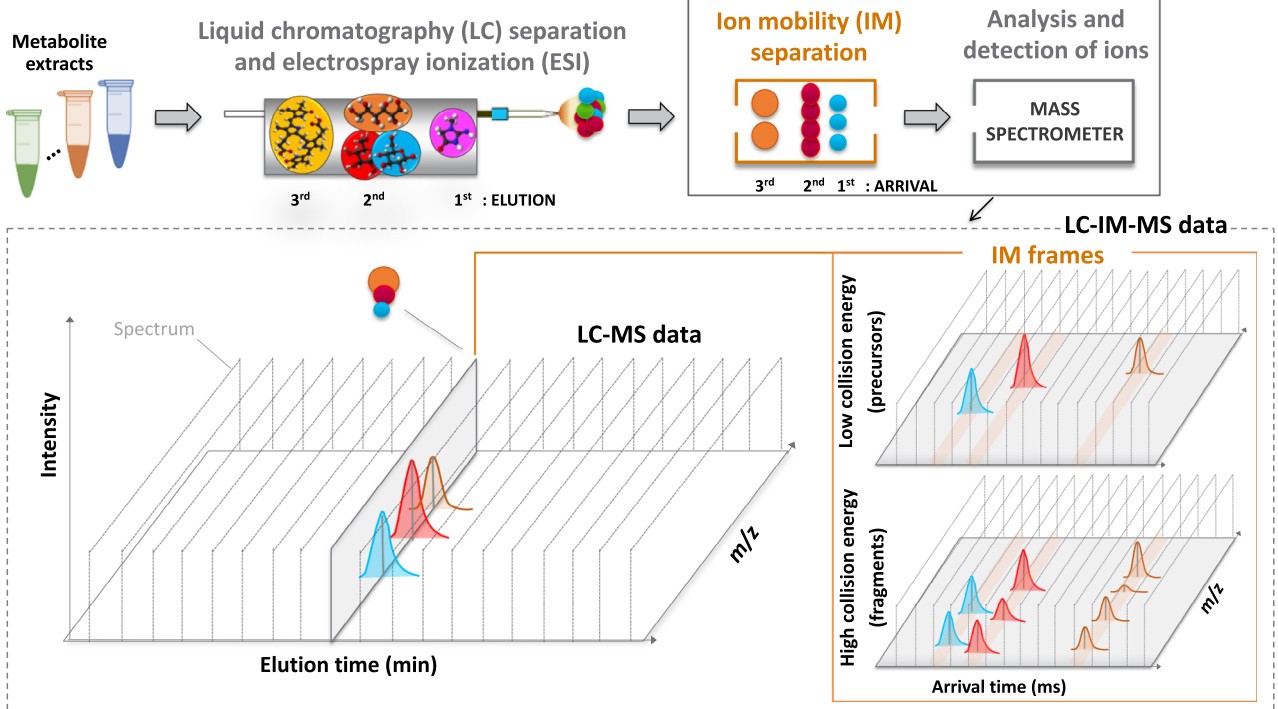

**Fig. 1 | Analytical workflow for multidimensional metabolite profiling by LC-IM-MS and data structure.** Metabolite extracts are separated by LC, followed by IM, and analyzed by MS in the All-Ions DIA mode which alternates between low and high collision energies to capture precursor and fragment ion spectra within the same run. Spectra are represented by gray dashed lines. Rather than collecting a single spectrum at every LC time point, coeluting ions (i.e., with close elution times) in this example at the 2nd order of elution and represented by spheres and peaks, in blue, red and orange colors, could be further distinguished by the ion mobility separation where multiple spectra are collected into IM frames. Fragments are detected within the same elution and mobility time window as their precursors. Figure adapted from previous work[69], with permission from Elsevier.

led to a total acquisition time of 9 min per run. Compared to the methods typically used to perform GC-MS-based global metabolomics[24], this LC method provides a ~3x faster sample analysis time and can detect other molecules which are undetectable by GC such as CoAs. DDA methods with short LC separation (<15 min) would be limited to only select the top 3–5 ions[18,26] per cycle to preserve the MS1 sampling rate and quantitation dynamic range, which in turn would result in MS/MS under sampling of medium-low-abundance ions. Therefore, after initial optimization the same LC system was utilized to perform the LC-IM-MS analyses in the All-Ions DIA mode (Fig. 1). A library with RT, CCS, and transitions (hereafter referred to as precursor and fragments) was built from the analysis of standards in deprotonated ion form. The list of metabolites can be found in Supplementary Table 1 and the library can be found in Supplementary Data 1. To evaluate the LC-IM-MS system against the gold standard SRM platform, dilution experiments were performed using representative standards. Supplementary Fig. 2 shows the calibration curves with linearity and increased sensitivity of LC-IM-MS over SRM for concentrations to as low as 0.075 pmol and covering four orders of magnitude. Next, 81 microbial samples from the various ABF-engineered hosts and conditions were analyzed by LC-IM-MS and processed using PeakDecoder.

## Developing the PeakDecoder algorithm

We implemented an alternative scoring algorithm for DIA metabolite identification which uses a 'raw spectrum centric' approach with UFD for ML training and a 'metabolite centric' approach with TDX for metabolite scoring (i.e., ML inference). Using only the unannotated LC-IM-MS DIA experimental spectra from biological samples, PeakDecoder learns to discriminate true co-elution (and co-mobility) of a precursor and its fragments from poor co-elution undistinguishable from random chance. As Fig. 2a shows, the PeakDecoder workflow has six steps for ML training and inference. First, the LC-IM-MS DIA data from the biological samples is processed in UFD mode using MS-DIAL[11]. Second, a preliminary training set is generated by using the detected and deconvoluted peak-groups as targets and producing their corresponding decoys. Third, TDX is performed using Skyline[14] to extract the precursor and fragment ion signals for the training set from all the LC-IM-MS DIA runs and export their XIC metrics. Fourth, a final training set is generated applying filtering for high-quality fragments to keep high-quality peak-groups as targets (i.e., precursor S/N > 20, and at least 2 fragments with mass error <15 ppm, RT difference to their precursor <0.1 min, and FWHM difference to their precursor larger than 2x the precursor FWHM; details in "Methods") and their corresponding decoys. A support vector machine (SVM) classifier is trained using multiple scores calculated from the XIC metrics of each peak-group in the training set: the cosine similarity between the expected and XIC intensities, and the mean and standard deviation of each precursor and its fragments for RT, LC-FWHM and mass error metrics (details in "Methods"). These scores are used as ML features which measure co-elution and similarity to the expected values[27]. After scoring the training set, the true and false positives can be used to estimate an FDR. Fifth, TDX is performed to extract the signals of the query set of metabolites in the library from all the LC-IM-MS runs and export their XIC metrics. Finally, the trained model is used to score the query set of metabolites and results can be filtered using the PeakDecoder score corresponding to the estimated FDR threshold from a table with pairs of values (FDR, PeakDecoder score) automatically generated.

PeakDecoder takes advantage of DIA spectra, where the combination of precursor and its fragments enable selective and sensitive detection of a molecule by a peak-group of co-eluting fragment ion

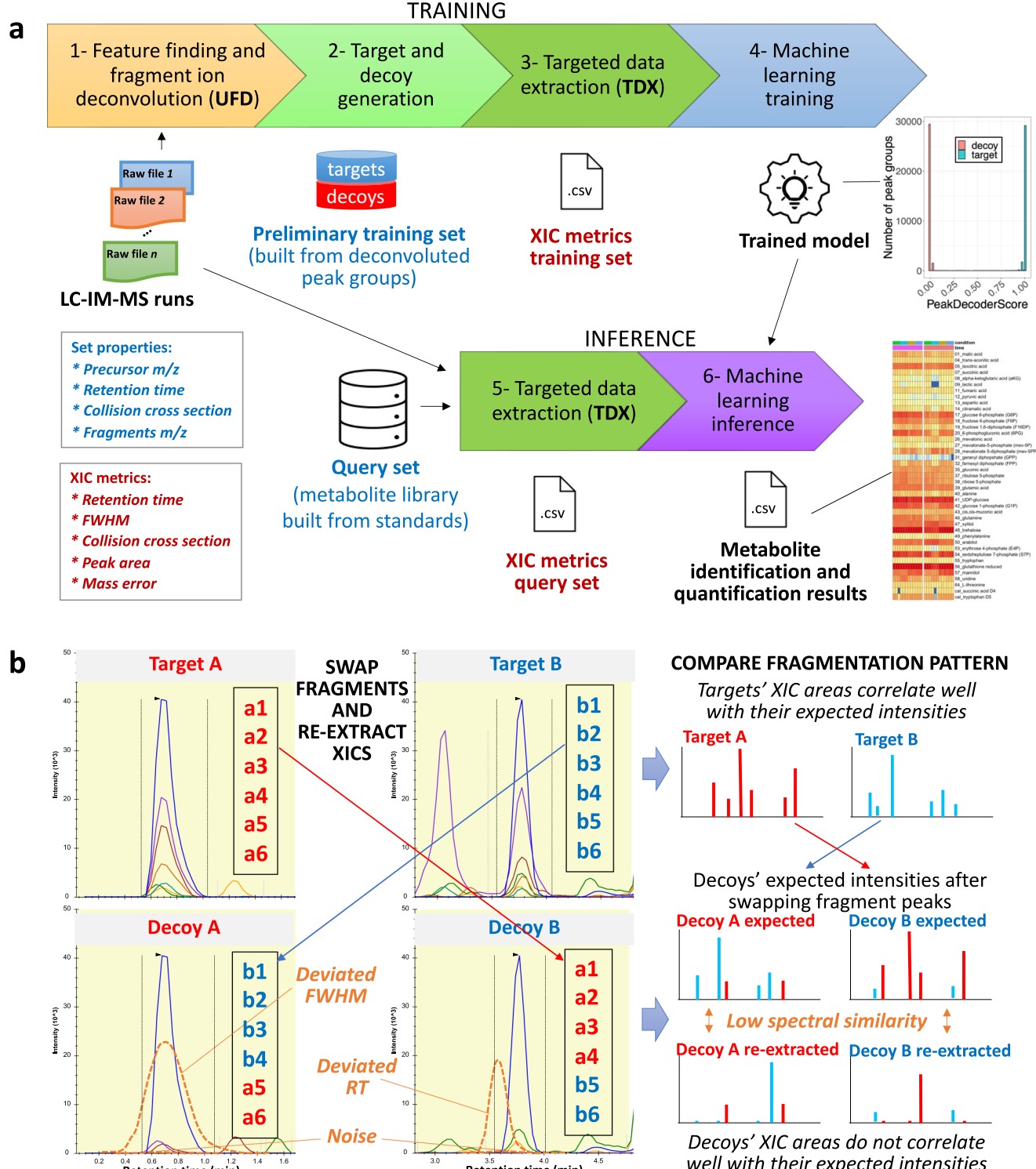

chromatograms[28]. Our algorithm is similar to the mProphet scoring method in terms of using decoy transitions[29,30]. The mProphet method introduced the concept of decoy transitions at the measurement level for SRM proteomics, and it was later adapted at the data extraction level for DIA. The decoy transitions are used to optimize a combination of the available individual scores and to derive statistical error rate estimates by parameterizing a null distribution. However, decoys in those original methods are generated from the protein database by reversing or shuffling the sequences. Due to the much larger structural diversity, more complex fragmentation mechanism and ubiquitous isomers compared to peptides, such decoy generation methods cannot be applied for small molecules. In contrast, PeakDecoder generates the decoys from the high-quality peak-groups deconvoluted from the LC-IM-MS DIA experimental spectra of the biological samples.

Methods to generate decoys from experimental spectra have been previously reported, however, from a DDA MS/MS target library (i.e., annotated spectra), first in proteomics[31,32] and more recently in metabolomics[20,22]. We propose an alternative strategy to generate decoys taking advantage of the comprehensive nature of the DIA spectra. Instead of generating decoys from the target library, we perform UFD and TDX in the LC-IM-MS DIA data to generate a

**Fig. 2 | Computational workflow for multidimensional metabolite profiling by LC-IM-MS. a** PeakDecoder algorithm. Step-1: data is processed in untargeted mode (UFD, MS-DIAL) to extract all precursor ion features (MS1) and their respective deconvoluted fragment ions (pseudo MS2) based on co-elution and co-mobility. Step-2: a preliminary training set is generated by using the detected and deconvoluted peak-groups as targets and producing their corresponding decoys. Step-3: targeted data extraction is performed (TDX, Skyline) to extract the precursor and fragment ion signals for the training set from all the LC-IM-MS runs and export their XIC metrics. Step-4: an SVM classifier is trained using multiple scores calculated from the XIC metrics of the training set. Before training, filtering for high-quality fragments is applied to keep high-quality peak-groups as targets (i.e., based on various thresholds for metrics of precursor and at least 3 fragments; details in "Methods") and their corresponding decoys in the final training set. The model

learns to distinguish true and false co-elution and co-mobility, independently of the features' metabolite identity. Step-5: TDX is performed to extract the signals of the query set of metabolites in the library from all the LC-IM-MS runs and export their XIC metrics. Step-6: the trained model is used to determine the PeakDecoder score of the query set of metabolites and estimate an FDR. **b** Example of decoy generation. The detected and deconvoluted peak-groups are associated by pairs and used as targets. For each pair of targets, A and B (fragments represented in red and blue colors, respectively), a pair of decoys is generated by keeping the same precursor and its properties and swapping the *m/z* values of 40–60% of the fragments (from the 6 most intense in this example). XIC metrics for targets correlate well with expected values but deviations and low spectral similarity occur for decoys (examples indicated in orange).

training set of peak-groups. The high-quality peak-groups constituted by the detected precursors (MS1) and its deconvoluted fragments (i.e., pseudo MS2) are used as targets. This strategy provides a noise-filtered 'clean' set of targets which was reported to be necessary to reach accurate estimates in spectrum level decoy-based methods[20]. We then employ a pairing and swapping strategy, similar to the precursor-swap method proposed by Cheng et al.[32], but rather than swapping precursors, we generate the respective decoy precursors and fragments from the same targets by swapping pairs of fragment *m/z* (Fig. 2b). Pairing precursors with the same number of fragments was used as an approach to increase the chances that the molecules are similar and to ensure that the overall distributions of general properties of targets and decoys are the same. Generating decoys by pooling and randomly adding fragments was avoided because it has previously shown poor performance (naive method)[20], as it increases the probability of generating unrealistic decoys. Since the deconvoluted data represent real molecules, our decoy strategy is valid in practice and the generated decoys comply with several conditions or properties, previously proposed for proteomics[31], to calculate FDR with a valid target-decoy model: (i) the decoy library has the same precursor *m/z* and charge distributions as the target library, (ii) target and decoy spectra include the same number of peaks and have the same intensity sum distribution, and (iii) decoy spectrum peaks are positioned on realistic *m/z* values (fragments that naturally occur).

Contrary to previously proposed methods for FDR assessment that rely on large libraries of annotated MS/MS spectra, PeakDecoder was designed to confidently identify metabolites from libraries, but independently of the number of metabolites in the library. The estimated error rates are independent of any library and therefore experimental or in silico generated libraries of any size could be potentially utilized. The scoring becomes 'metabolite centric' and provides the probability that a given metabolite is present in the sample based on the quality of its detected signals in the LC-IM-MS DIA data. After the model is trained directly from the unannotated LC-IM-MS DIA data, it can be used to automatically score metabolites in libraries.

Since PeakDecoder generates the decoys from unannotated LC-IM-MS DIA experimental spectra, the size of the target library does not affect its performance. However, the performance of PeakDecoder depends on the training set and the validity of the estimated FDR depends on the number of generated false positives. The size and quality of the training set can be controlled in two ways: the parameters of the UFD tool used to generate the preliminary training set (Fig. 2a, Step-1) and the filtering for high-quality fragments used to generate the final training set (Fig. 2a, Step-4). At the same time, a tradeoff in the quality of peak-groups is necessary to avoid overfitting and perfect training accuracy, and thus, to estimate a reliable FDR. These components allow the user to define the quality of the resulting annotations and are evaluated using microbial data in the next section.

## Applying PeakDecoder in microbial samples

We processed the microbial LC-IM-MS data using PeakDecoder. The datasets represented varied sample complexity and feature density: low for *A. pseudoterreus* & *A. niger*, medium for *P. putida,* and high for *R. toruloides*. Supplementary Fig. 3 shows the distributions of ions illustrating the general properties of the targets and decoys generated for training. Figure 3 shows results for the *P. putida* samples. The PeakDecoder score which combines individual scores provided an improved discrimination power between targets and decoys (Fig. 3a). An example of chromatograms and filtered IM window for 'fructose 1,6-diphosphate (F16DP)' from the standard (precursor *m/z* 338.98877, RT 4.95 min, CCS 155.00 and 6 fragments) and a microbial sample is shown in Fig. 3b, confidently identified with a PeakDecoder score of 0.9966 and 0.005 q-value. Supplementary Fig. 4 shows the PeakDecoder training performance for all microbial samples and a summary is shown in Table 1. A total of 2683 features could be confidently annotated. Annotations could be attributed to either all dimensions by RT-CCS-DIA or to RT-CCS for features without detected fragments (i.e., MS1 level only). The number of features annotated in each dataset includes replicates and is independent of the number of unique metabolites identified. For instance, in the case of the *A. pseudoterreus* & *A. niger* dataset, many more features were annotated, indicating that metabolites were detected in multiple replicates across all sample conditions.

To control the size and quality of the final training set, we defined the parameters of the UFD tool (Fig. 2a, Step-1) and the filtering for high-quality fragments (Fig. 2a, Step-4) according to the characteristics of our analytical method and instrumentation (e.g., fragments with RT difference to their precursor <0.1 min) and annotation quality preferred (e.g., at least 3 fragments). Because of the low sample complexity of the *A. pseudoterreus* & *A. niger* dataset, a smaller number of deconvoluted peak-groups were detected, therefore only 234 target peaks could be generated for training and were not sufficient for a good FDR estimation. The medium sample complexity of the *P. putida* dataset provided the best FDR estimation. Supplementary Fig. 5 shows that the training performance was not significantly impacted by the deconvolution parameters if the numbers of targets was sufficient (accuracy >98.86 if the resulting training set contained between 2760 and 6720 targets), but at the same time, if the classifier resulted in a close-to-perfect accuracy (>99), the minimum non-zero FDR that could be estimated was affected due to the small number of false positives. Conversely, the high sample complexity in the *R. toruloides* dataset resulted in poor performance when using the default filtering for high-quality fragments generating 8674 targets/decoys for training, where the minimum non-zero estimated FDR for the highest PeakDecoder score was 3% (Supplementary Fig. 6a). Stringent values were applied to filter the high-quality fragments generating 1400 targets/decoys (Supplementary Fig. 6b) and a minimum non-zero estimated FDR of 1% could be obtained. The results indicate that more training data does not translate into higher accuracy and further improvements for filtering high-quality fragments (i.e., generating a smaller training set

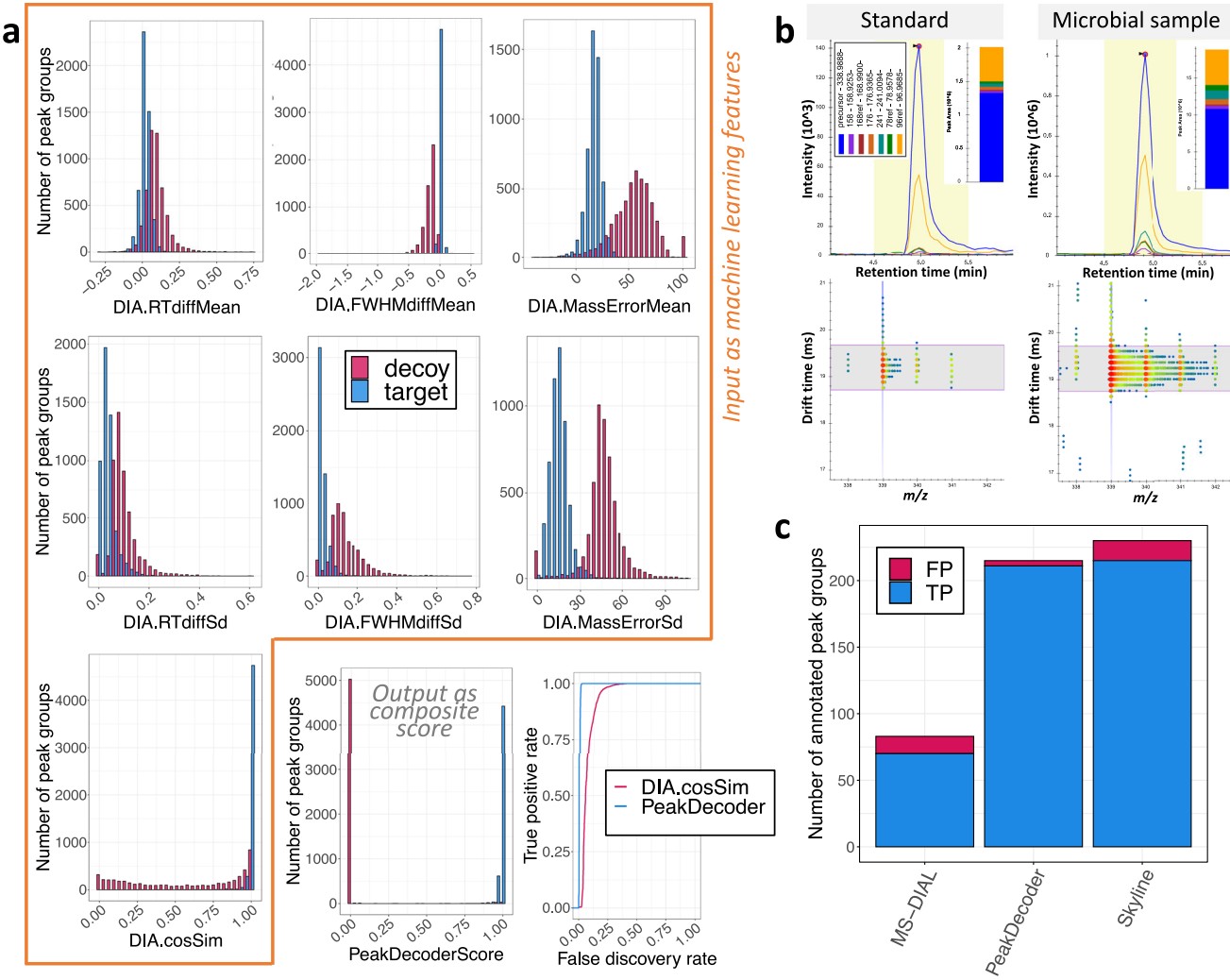

**Fig. 3 | Analysis of microbial samples by LC-IM-MS using PeakDecoder.**
**a** Comparison of scores in training. Targets and decoys are represented by blue and red colors, respectively. Distributions of LC-IM-MS peak-groups by each individual score (highlighted in orange) showed limited separation of targets and decoys. Individual scores used as machine learning features were combined into the composite PeakDecoder score providing an improved separation power and resulted in a larger number of true positives for lower FDR thresholds than the cosine similarity score, which is the best score individually. **b** Example of chromatograms and filtered ion mobility window. Signals for 'fructose 1,6-diphosphate (F16DP)' from the standard (precursor *m/z* 338.98877, RT 4.95 min, CCS 155.00 and 6 fragments) and corresponding peaks from a microbial sample (annotated by

PeakDecoder). Chromatograms show the same relative abundances in the standard and the microbial sample confirming the correct metabolite annotation based on fragmentation pattern and RT. The IM frame at the LC apex shows the filtering window corresponding to the expected CCS and highlights the precursor with multiple isotopic peaks. **c** Benchmarking of identification performance compared to manual curation. True positives (TP) and false positives (FP) are represented by blue and red colors, respectively. PeakDecoder at 1% estimated FDR increased TP annotations (211) compared to MS-DIAL (TP = 70, total score > 60) and decreased by 4 compared to Skyline (TP = 215, cosine similarity > 0.8), while decreasing FP annotations (FP: PeakDecoder = 4, MS-DIAL = 13, Skyline = 15). Results from the *P. putida* samples (*n* = 22). Source data are provided as a Source data file.

with peak-groups of appropriate quality) could be needed for datasets with high sample complexity. Supplementary Fig. 6c depicts results from a deuterated standard (tryptophan d5) spiked in solvents and in a microbial sample matrix.

In all samples, PeakDecoder could identify the metabolites expected in at least one condition of each microbial dataset (a list of unique metabolites generated by manual inspection of the most intense replicates). A handful of cases that were missed initially were recovered after manual correction of the Skyline chromatogram peak detection. Supplementary Tables 2–4 show the scores and annotation confidence level (best replicate per metabolite). In addition, we performed targeted analyses of a subset of metabolites by SRM in *P. putida* samples and GC-MS analyses of *A. pseudoterreus* and *A. niger* samples to further evaluate the performance of our method in biological samples (Supplementary Fig. 7). Similar trends were observed for the metabolites identified in common by the different platforms.

## Comparing PeakDecoder to other workflows

UFD (MS-DIAL) and TDX (Skyline) are two different approaches with different advantages and disadvantages. While the UFD does not rely on a library and high-quality peak-groups from its deconvolution results can be used for training, applying TDX offers advantages over UFD for annotation in DIA, particularly for All-Ions data, where the full mass range is co-fragmented, and the likelihood of interference greatly increases as sample complexity increases. In complex samples, multiple precursors with very similar RT and DT are present as a series of partial overlapping ions which compromise the effectiveness of UFD algorithms. However, when performing TDX, only the relevant chromatograms are extracted in a directed and highly selective fashion.

PeakDecoder combines both UFD and TDX strategies and addresses limitations in the respective existing tools. Specifically, the re-extraction of signals by TDX in Skyline allows specifying a DT offset for fragments characteristic for the IM instrumentation[33]

**Table 1 | Summary of PeakDecoder training performance and identification results in microbial samples**

| Dataset (number of runs) | Collision energy | # Target peak-groups in training | Training accuracy (average 10-fold cross-validation) | # Unique metabolites | | # Annotated features | |
|---|---|---|---|---|---|---|---|
| | | | | RT-CCS | RT-CCS-DIA | RT-CCS | RT-CCS-DIA |
| A. pseudoterreus & A. niger (46) | 20 V | 234 | 98.50 | 12 | 27 | 322 | 909 |
| P. putida (22) | 20 V & 40 V | 5152 | 98.86 | 12 | 25 | 329 | 239 |
| R. toruloides (48) | 20 V & 40 V | 1400 | 97.04 | 14 | 22 | 636 | 248 |

Number of unique metabolites and number of annotated features refer to the identifications obtained by matching against our library of 64 metabolites by accurate mass plus either 2 dimensions with RT-CSS (annotations at the MS1 level only, i.e., no detected fragments) or 3 dimensions with RT-CCS-DIA (i.e., including fragments). Unique metabolites do not count repetitions. Annotated features include sample replicates and different conditions. Data for the A. pseudoterreus and A. niger samples were acquired with 20 V collision energy (CE) and for P. putida and R. toruloides with both 20 V and 40 V CE. The number of annotations varies per dataset due to CEs and their different number of conditions and replicates.

(see "Methods"), which is not applied in MS-DIAL and results in a poor deconvolution of fragments with the smaller masses. On the other hand, the UFD in MS-DIAL allows accurate CCS evaluation using the experimental CCS values, which is not available in Skyline because it does not perform a peak detection in the IM dimension and is limited to use the CCS information as a filtering window (e.g., Fig. 3b). Besides combining the best features of these two tools, PeakDecoder uses the peak shape metrics, combines the individual scores into a composite score, and allows FDR estimation, all of which are impossible with MS-DIAL or Skyline alone.

To benchmark PeakDecoder and evaluate the reliability of our FDR estimation, we performed a comparison against the ground truth generated from manually curating the full *P. putida* dataset, with 550 peak-groups including 233 positives and 317 negatives (Fig. 3c). Due to the poor deconvolution of fragments with the smaller masses, MS-DIAL resulted in the lowest number of true positives (TP = 70), even when using a relaxed threshold for its total score (>60). While Peak-Decoder at 1% estimated FDR missed 4 TP compared to Skyline (cosine similarity >0.8), it decreased the number of false positive annotations (FP: PeakDecoder = 4, MS-DIAL = 13, Skyline = 15). The estimated 1% FDR corresponded to a ~2% actual FDR, and while there is an under-estimation and the results are limited by our small library, they show that PeakDecoder could be used to filter out FP.

Our decoy strategy for DIA data together with IM and LC conveys a powerful multidimensional characterization of metabolites that address several important challenges. For many metabolites only a few characteristic fragment ions can be detected, rendering the use of classic spectral similarity searches unreliable[18]. Moreover, some metabolites could not be detected with even a single fragment. In these cases, the CCS increased the identification confidence compared to using the RT and accurate mass alone. Since our library was built from pure standards, even for standards without fragments, the identification based on RT and CCS could be considered as a con-fidence of "Level 1" according to the Metabolomics Standards Initiative[34], as they are two different analytical techniques. Besides, multidimensional LC-IM-MS increases the separation, important for metabolites that co-elute, where DIA alone is challenged by fragments common to co-eluting metabolites. Figure 4 illustrates the power of multidimensional separations to increase the selectivity and therefore increase the annotation confidence and quantitation accuracy. The number of possible LC-IM-MS peaks from MS-DIAL untargeted feature detection results matched within tolerances (0.01 mass, 0.2 min RT, and 0.8% CCS) was reduced when using all dimensions. High IM resolving power is essential for small molecules and current IM instruments are able to separate CCS differences as low as 0.8%.

### Metabolomics of *A. pseudoterreus* and *A. niger* strains

PeakDecoder was applied for metabolomics profiling of *A. pseudo-terreus* and *A. niger* strains engineered to produce 3-hydroxypropanoic acid (3HP) using the β-alanine pathway[35]. Our three engineered *A. pseudoterreus* strains[24] with varying levels of 3HP production (low, medium, and high) and their parent strain (ATCC 32359 Δ*cad: cis-aconitic acid decarboxylase deletion*) were analyzed. Since the engi-neered *A. pseudoterreus* strains produced significant amount of other organic acids[24], we also developed and profiled *A. niger* strains engi-neered with the same β-alanine pathway. Five engineered *A. niger* strains exhibiting different levels of 3HP production (low, medium, high, higher, and highest) and their parent strain (ATCC 11414) were included.

Metabolomics profiling of 3HP-producing *A. pseudoterreus* and *A. niger* strains revealed species-specific metabolic responses to increasing 3HP production. Specifically, we found that L-aspartate, the precursor to the β-alanine 3HP production pathway, showed very little change in 3HP producing *A. pseudoterreus* strains, while its level decreased significantly in 3HP producing *A. niger* strains (Fig. 5). In the

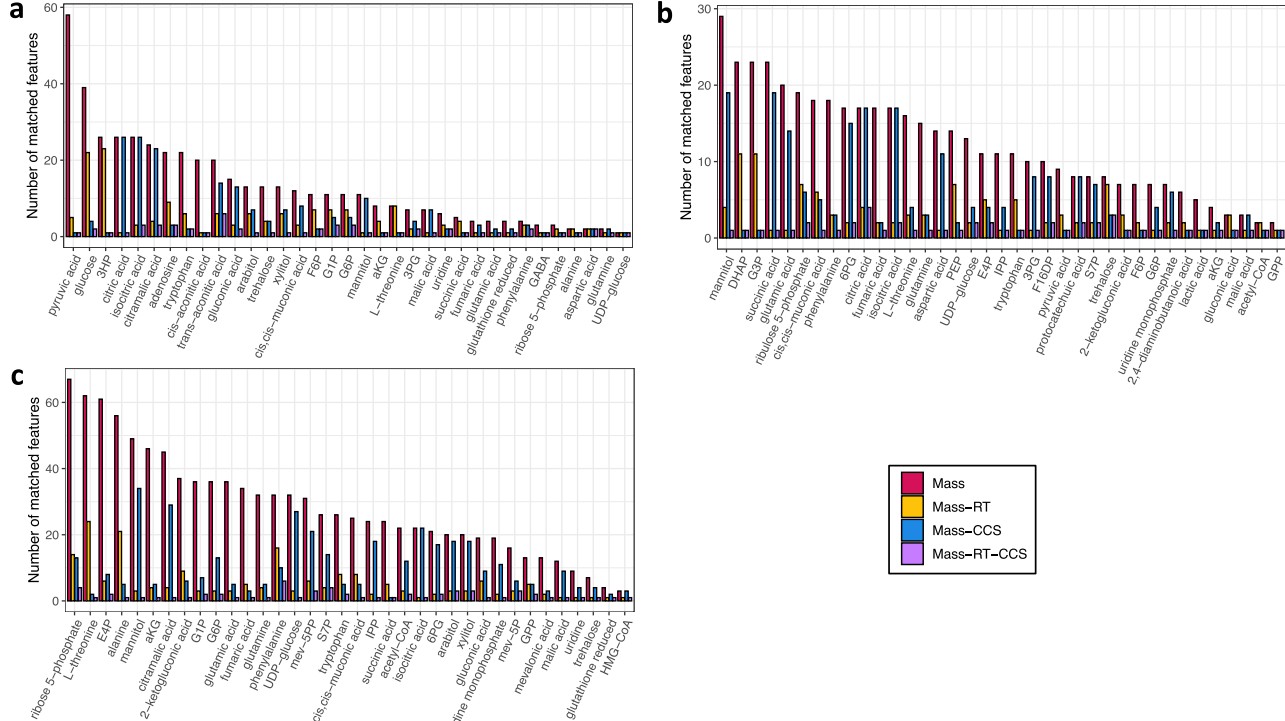

**Fig. 4 | Annotation selectivity by different analytical separations in microbial samples. a** *A. pseudoterreus* and *A. niger* (*n* = 46). **b** *P. putida* (*n* = 22). **c** *R. toruloides* (*n* = 48). Bars represent the number of possible LC-IM-MS peaks from untargeted feature detection results matched within tolerances. Colors represent the type of match: red=Mass, yellow = Mass-RT, blue = Mass-CCS, and purple = Mass-RT-CCS. In all three microbial datasets, using accurate mass alone resulted in the highest number of features, notably for the metabolites with smaller masses. Combining

accurate mass to either RT or CCS reduced the number of matched features. By combining accurate mass with both RT and CCS, the number of possible features was reduced to one in most cases. These results illustrate the power of multi-dimensional separations to increase the annotation confidence and quantitation accuracy in metabolomics studies by resolving the high degree of structural diversity derived from isomers and isobars. Source data are provided as a Source data file.

β-alanine 3HP production pathway, L-aspartate is converted to 3HP via β-alanine using multiple aminotransferases.

L-aspartate → β-alanine + $CO_2$

β-alanine + pyruvate → malonate semialdehyde + L-alanine

L-alanine + α-ketoglutarate → pyruvate + L-glutamate

oxaloacetate + L-glutamate → L-aspartate + α-ketoglutarate

(net reaction) β-alanine + oxaloacetate → malonate semialdehyde + L-aspartate

malonate semialdehyde + NADPH + $H^+$ → 3HP + $NADP^+$

L-aspartate is first converted to β-alanine by aspartate-1-dec-arboxylase, and the amino group from β-alanine is transferred to pyruvate yielding malonate semialdehyde and L-alanine by β-alanine/pyruvate aminotransferase. Malonate semialdehyde is converted to the final product 3HP by 3-hydroxypropionate dehydrogenase, but L-alanine needs be converted back to pyruvate by alanine transaminase by transferring the amino group to α-ketoglutarate generating L-glutamate. The aminotransferase cycle can be closed by generating the precursor L-aspartate by transferring the amino group from L-glutamate to oxaloacetate. Therefore, the amino group acceptor and donor pairs (pyruvate/L-alanine and α-ketoglutarate/L-glutamate) play an important role in the β-alanine 3HP production pathway.

PeakDecoder allowed us to investigate the changes in these amino group acceptor and donor pairs as well as undesired byproducts such as 4-aminobutyric acid and 2,4-aminobutanoic acid. Similar to the conversion of L-aspartate to 3HP via β-alanine, L-glutamate can be converted to succinate via 4-aminobutyric acid (GABA). In the GABA degradation pathway, GABA is first deaminated to succinate semi-aldehyde by 4-aminobutyrate aminotransferase UGA1 using α-keto-glutarate/L-glutamate pair, which was one of the most significantly upregulated enzymes in the engineered *A. pseudoterreus* strains in our

previous study[24]. In this study, we observed significantly decreased levels of succinate semialdehyde in *A. niger* strains producing high levels of 3HP using the developed workflow, confirming that the engineered 3HP pathway is affecting the GABA degradation pathway in *A. niger* as well. We also previously hypothesized that the promiscuous activity of upregulated UGA1 resulted in the accumulation of 2,4-dia-minobutyric acid from L-aspartate via L-aspartate 4-semialdehyde in *A. pseudoterreus*. Here, we found that the accumulation of 2,4-diamino-butyric acid was not consistently observed in the engineered *A. niger* strains in contrast to the observation in *A. pseudoterreus*. This is likely due to the significantly decreased level of the precursor L-aspartate in the engineered *A. niger* strains. The level of L-aspartate 4-semialdehyde is consistently lower in the engineered *A. pseudoterreus* strains, but not in the engineered *A. niger* strains.

### Omics of engineered muconate-catabolizing *P. putida* strains

*P. putida* has biochemical properties that make it ideal for hosting biochemical transformations[36]. Due to its naturally diverse and flexible catabolism it can metabolize aliphatic, aromatic, and heterocyclic compounds in addition to glucose[37]. To use *P. putida* for industrial bioprocessing, genetic modifications must be incorporated into the strains requiring the expression of heterologous genes and pathways. Chaves et al. studied the importance of chromosomal integration location, which affects heterologous protein expression independent of typical design parameters such as copy number, promoter, and terminator type[37]. Wild-type (WT) *P. putida* KT2440 cannot grow on cis,cis-muconate as a sole carbon source, despite using this compound as a key intermediate in aromatic catabolism. To enable muconate catabolism, a transmembrane transporter for muconate (mucK) was integrated into three different chromosomal sites (PP2224, PP1642,

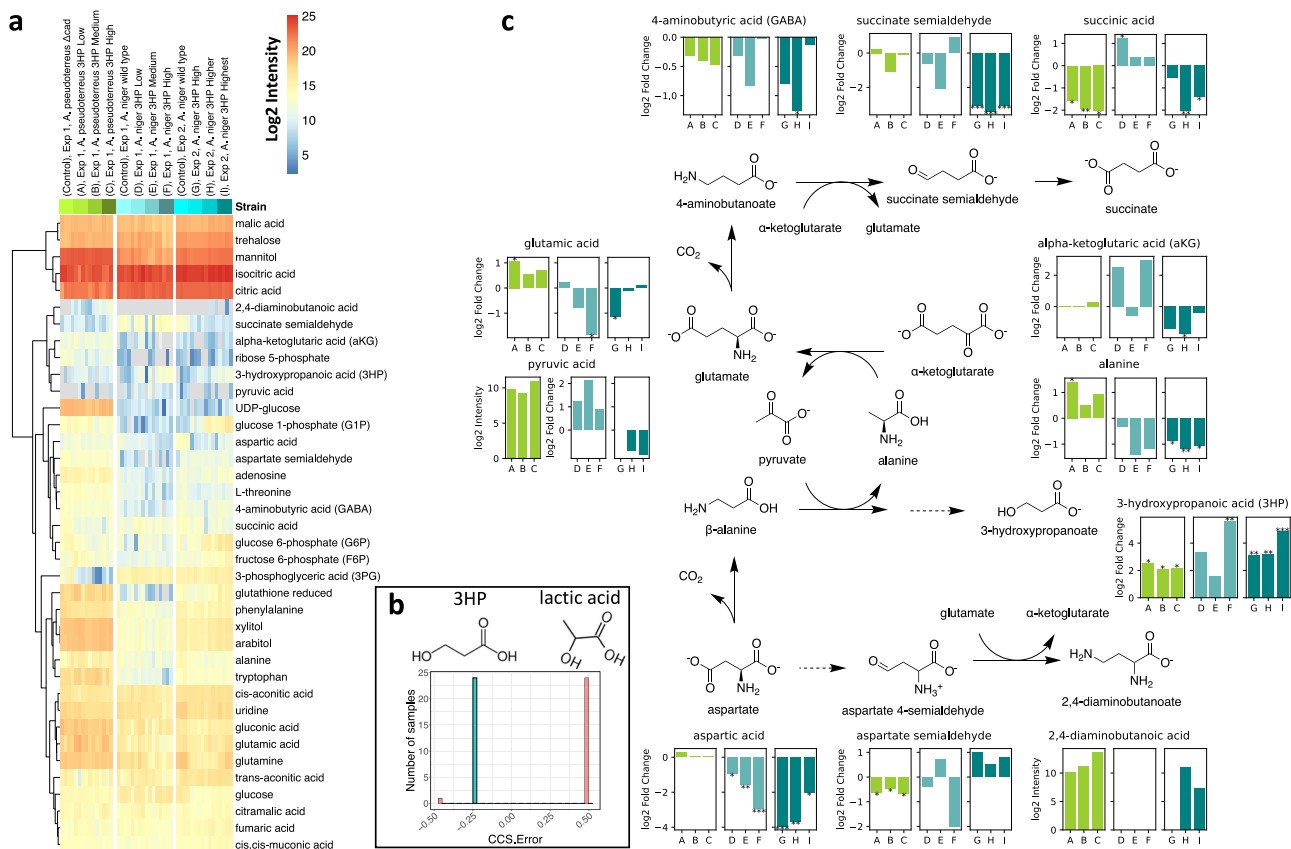

**Fig. 5 | Metabolomics profiling of 3HP-producing *A. pseudoterreus* and *A. niger* strains. a** Relative and label-free intracellular metabolites levels quantified by PeakDecoder (*n* = 46). Red, yellow, and blue colors indicate high, medium, and low log2 intensity values, and gray color indicates missing values. **b** CCS errors of the good-quality features in 24 samples confirmed the detection of 3HP (green bar, 113.8 CCS) instead of lactic acid (orange bar, 113.0 CCS), which is an isomeric molecule (same formula but with different 3D structure). **c** Metabolites in the 3HP production pathway and their log2 fold changes over the control sample (parent strain). Statistical analysis was performed using the IMD-ANOVA method. Stars indicate statistically significant changes (*p-value <0.05, **p-value <0.01, and ***p-value <0.001). Y-axis for pyruvic (*A. pseudoterreus*) and 2,4-diaminobutanoic acids represent mean log2 intensity due to no detection in the control strain. Source data are provided as a Source data file.

and PP5042). Samples with the transmembrane transporter were grown in M9 minimal medium supplemented with 30 mM muconate and analyzed using a targeted proteomics approach to quantify the amount of mucK present. The previous results showed that the growth rate with muconate inversely correlated with the expression levels of the transporter.

To provide additional insights into the metabolic changes during growth with a new carbon substrate, we used PeakDecoder to quantify metabolites in *P. putida* WT grown on glucose and mucK (PP2224, PP1642, and PP5042) grown on muconate. Compared to WT, muconate-catabolizing *P. putida* mucK strains showed decreased levels of metabolites in the ED-EMP cycle (glucose utilization) such as fructose 6-phosphate, fructose 1,6-diphosphate, and glyceraldehyde 3-phosphate among others (Fig. 6). Targeted proteomics, performed on the same cell pellets of mucK strains, showed a corresponding decrease in enzymes that are part of ED-EMP pathway and in levels of pyruvate dehydrogenase and pyruvate carboxylase, which catalyze the conversion of pyruvate to acetyl-CoA and to oxaloacetate, respectively. These lower levels match with the accumulation of pyruvate in mucK strains cultured with muconate. Accumulation of pyruvate had also been observed in *P. putida* grown in a glucose:benzoate mixture vs glucose alone[38]. In contrast, increased levels of metabolites (α-ketoglutarate, fumarate, malate) and enzymes from the TCA cycle were observed in the mucK strains. Levels of enzymes at the entrance point of acetyl-CoA into TCA cycle and those routing succinyl-CoA and succinate into TCA cycle were upregulated in mucK compared to WT.

Muconate is metabolized via the β-ketoadipate pathway before joining the central carbon metabolism via acetyl-CoA and succinate. Although no metabolites in the beta-ketoadipate pathway were detected, enzymes in this pathway were significantly upregulated in the mucK strains compared to WT which is expected considering muconate was used as the carbon source. Changes in metabolite levels in peripheral pathways were also clear. Gluconate and 2-ketogluconate were lower or not detected in mucK compared to WT which is in line with the absence of glucose supplementation in the strains with the transporter. These results suggest a shift in metabolism supported on succinate and acetyl-CoA fueling the TCA cycle from the β-ketoadipate pathway and less reliance on ED-EMP pathway when muconate is used as carbon source. Similar results were observed when *P. putida* was grown in p-coumarate[39]. Supplementary Fig. 8 shows the targeted proteomics quantitation results and changes in metabolic pathways for mucK PP1642 and mucK PP2224 compared to the WT.

**Mevalonate pathway in *R. toruloides* strains**

*R. toruloides* is an important model microorganism for synthetic biology and industrial biotechnology due to its capacity to bioconvert lignin, the most underutilized component of plant biomass[40]. Metabolic engineering of *R. toruloides* can generate distinct bioproducts including bisabolene, the immediate precursor of bisabolane and an alternative to D2 diesel fuel[41]. For example, the Agile BioFoundry *R. toruloides* strain, GB2, can produce bisabolene in high quantities of 2.2 g/L from lignocellulosic hydrolysate in 2-L fermenters[42]. Another

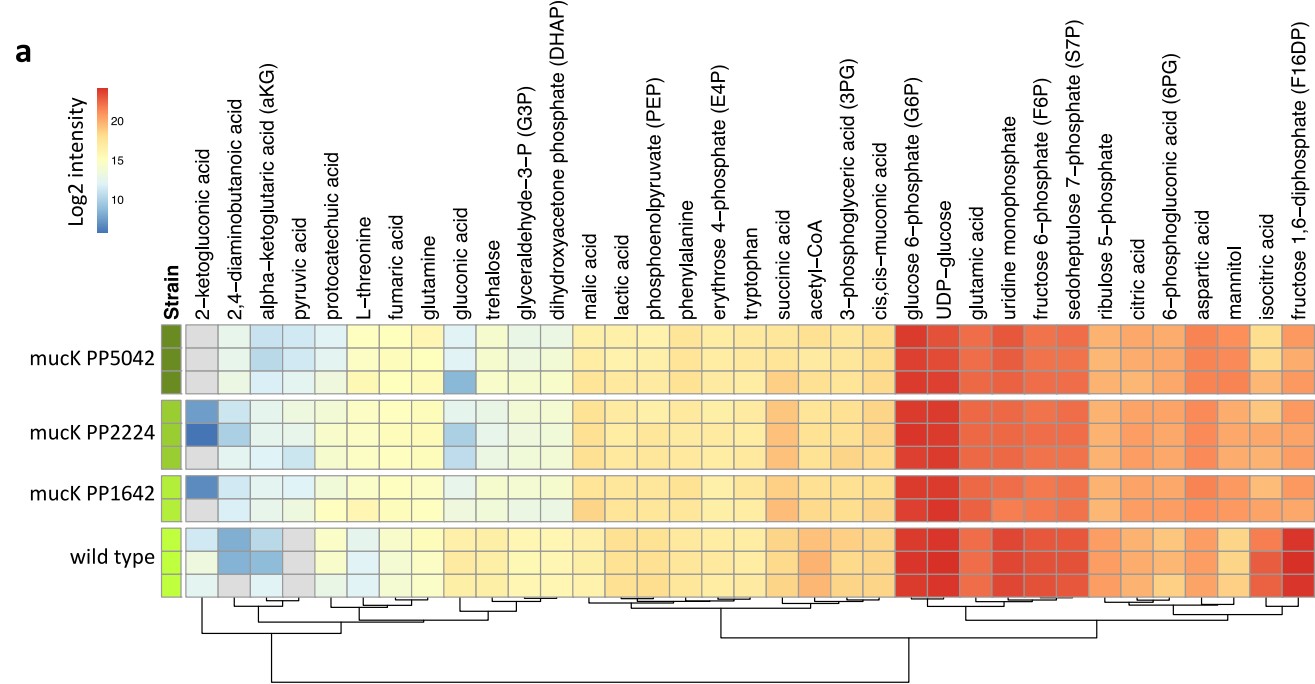

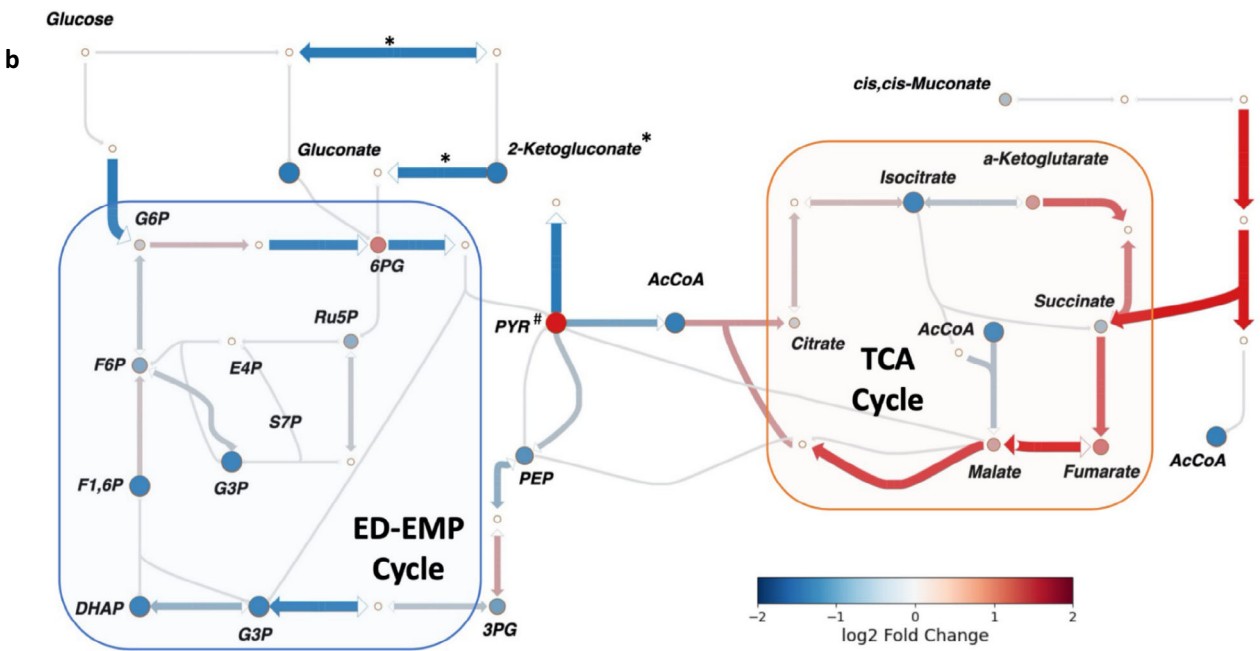

**Fig. 6 | Metabolomics and proteomics profiling of *P. putida* wild type and engineered muconate-catabolizing strains. a** Relative and label-free intracellular metabolites levels quantified by PeakDecoder (*n* = 22, with 11 samples and 2 collision energies per sample). Red, yellow, and blue colors indicate high, medium, and low log2 intensity values, and gray color indicates missing values. **b** Glucose and muconate catabolism pathways of mucK PP5042 and fold changes compared to the wild-type strain. Circles indicate metabolites and arrows indicate proteins detected by SRM. Symbols indicate protein detection: * detected in the wild type but not detected in the mucK samples and # detected in the mucK but not in the wild type. Source data are provided as a Source data file.

key advantage of *R. toruloides* is that it can grow on mixed-carbon sources and tolerate growth inhibitors often present in lignocellulosic hydrolysates[40]. However, these hydrolysates present a significant challenge in biochemical conversion due to feedstock variability[43].

We employed PeakDecoder and global proteomics analyses to characterize *R. toruloides* GB2 cultured on lignocellulosic hydrolysates derived from corn stover with variable levels of ash (A) and moisture (M), each parameter cataloged as High (H) or Low (L) and the 4

possible combinations of them (HAHM, HALM, LAHM, and LALM). Samples were collected at two time points of fermentation, during exponential growth (36 h) and at the end of this growth phase (60 h). A total of 37 unique metabolites were confidently detected in at least one sample and quantified across all samples (Supplementary Fig. 9).

Bisabolene is produced upon the introduction of bisabolene synthase and its precursor, farnesyl pyrophosphate (FPP), is part of the mevalonate pathway. Figure 7 details our mevalonate pathway

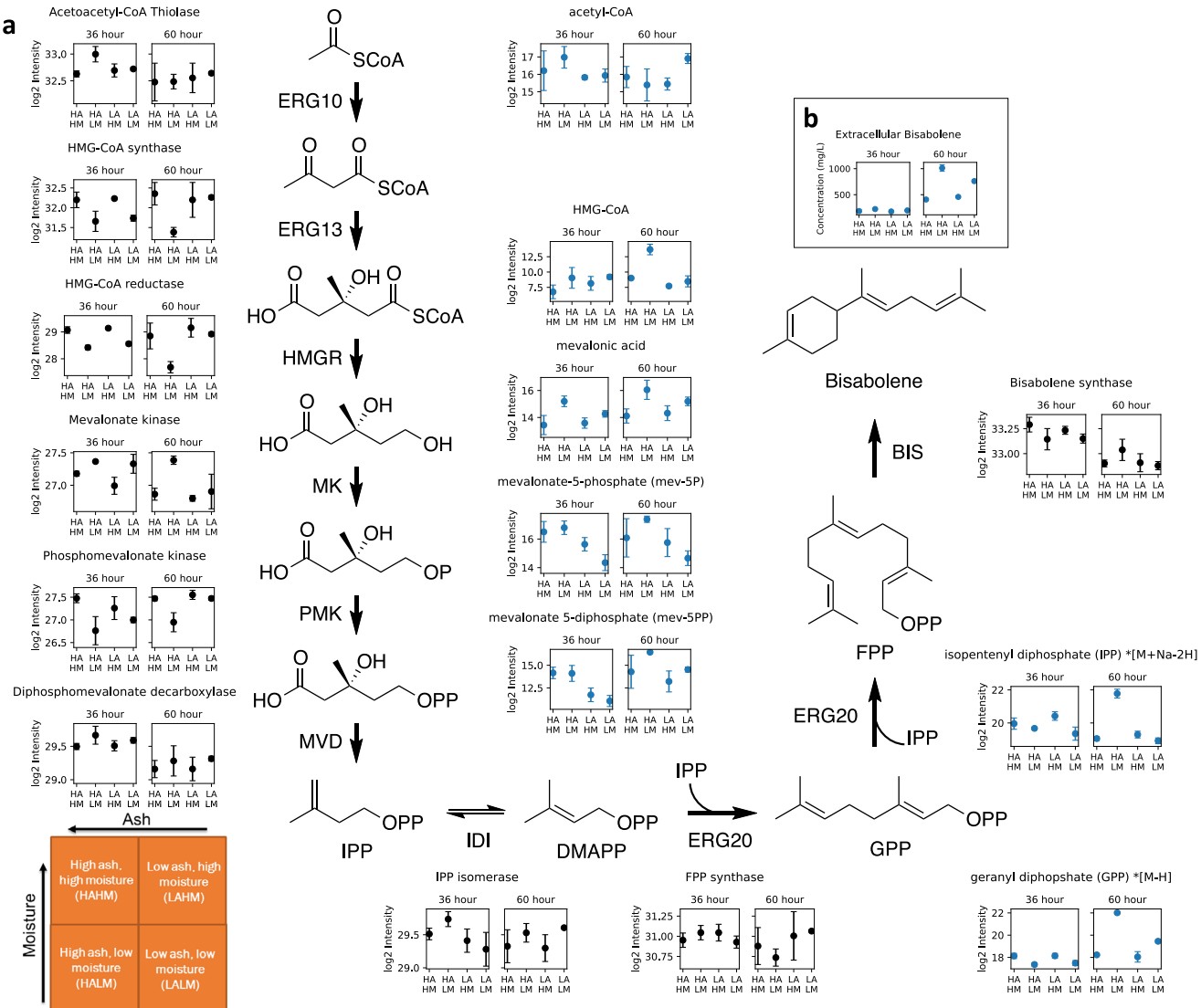

**Fig. 7 | Metabolite and enzyme levels in the mevalonate pathway of *R. toruloides* strains. a** Relative and label-free abundance levels are represented in blue for metabolomics (*n* = 48, with 24 samples and 2 collision energies per sample) and black for proteomics (*n* = 24 samples). Strains were grown in hydrolysates with different contents of ash and moisture and collected at 36 and 60 h. **b** Bisabolene production (extracellular) captured in a dodecane overlay. Data are presented as mean values with error bars from standard deviation of 3 biological replicates. Source data are provided as a Source data file.

metabolomic and proteomic results. The levels of HMG-CoA were significantly higher in cells cultured in high ash low moisture (HALM) conditions at 60 h. The rate limiting step in the mevalonate pathway is the conversion of HMG-CoA to mevalonic acid by 3-hydroxy-3-methylglutaryl-coenzyme A reductase (HMGR)[44]. *R. toruloides*, like mammalian systems, has only one HMGR gene[45] and mammalian HMGR and yeast Hmg2p (from *Saccharomyces cerevisiae*) are both subject to feedback control by the sterol pathway[46]. Previous studies identified FPP or FPP derivatives as the positive signal for HMGR degradation in yeast[44,46]. Here we detected isopentenyl pyrophosphate/dimethylallyl pyrophosphate (IPP/DMAPP) and geranyl pyrophosphate GPP. IPP and DMAPP are isomeric molecules which could not be separated by the current drift tube IM resolution or our LC method, and the sodiated adduct ion provided better quantification on the samples. We observed that the absolute differences in mass and CCS were larger for IPP/DMAPP and GPP ions compared to the rest of the identified molecules (respectively, −0.0858 and −0.0449 *m/z*, and 1.03 and 0.9 CCS). However, after adjusting these values we could quantify these molecules across all samples and observed a consistent trend. The levels of IPP/DMAPP, GPP (precursor to FPP), and extracellular FPP-derived bisabolene (Fig. 7) which were higher in cells grown on HALM hydrolysates at 60 hr compared to all other conditions, could explain the decreased level of HMGR detected in the proteomic analysis and the subsequent accumulation of HMG-CoA (i.e., at 60 h for the comparison of HALM vs HAHM, log2FC of HMGR: −1.16, log2FC of HMG-CoA: 4.67, respective *p*-values are $1.82 \times 10^{-3}$ and $3.28 \times 10^{-3}$; see Supplementary Table 5). Previously, it had been observed that when sterol pathway flux is high, degradation of HMGR is fast and its levels are low[46] and this is what was revealed for GB 2 growth on HALM hydrolysate after 60 h by our advanced analytical workflow.

## Discussion

Using synthetic biology applications, we have described and demonstrated an optimized analytical method, a chromatography method prediction tool, and an alternative metabolomics algorithm for robust processing of multidimensional data acquired in state-of-the-art LC-IM-MS instrumentation. The advantages of using LC-IM-MS with DIA and PeakDecoder enable high-throughput analyses with increased metabolite coverage and more confident annotation due to several

aspects: (1) in terms of data acquisition, our 9 min LC method is faster than the GC methods typically used, (2) the IM dimension further separates more analytes and increases annotation confidence by combining CCS and RT compared to LC alone, (3) DIA further increases annotation confidence with fragmentation information and provides better reproducibility and dynamic range than DDA, and (4) Our PeakDecoder score provides a confident metric for metabolite annotation. These tools have the potential to enable faster and more accurate testing of strains generated by high-throughput engineering workflows and therefore accelerate the DBTL cycle. Engineered microbes (e.g., bacteria, yeast, fungi) producing bioproducts (e.g., fuels, chemicals, materials) in a sustainable way are necessary to achieve a strong bioeconomy and decrease dependence on fossil fuels. Our analytical and computational workflow will provide capabilities for fast analysis of current and new metabolites of interest and is broadly useful, beyond the ABF consortium, in a wide range of environmental and biological metabolomics research.

Our multidimensional metabolite library built from 64 standards is available as a resource to the community and we expect it to be expanded, since the combination of RT and CCS with co-elution and co-mobility profiles from DIA fragmentation patterns significantly increases confidence in overall compound identification. Besides, DIA spectra are a permanent and comprehensive digital record of all detectable ions in the sample, which can be re-processed as new libraries or new tools become available, and without the need of re-analyzing the samples for acquiring new data. Consequently, new biochemical hypothesis could be investigated using the existing microbial data, and DIA would allow evaluating/investigating side effects such as undesired pathways activated or undesired products, which would be missed with targeted-MS methods.

Although more developments could be explored, for example, other decoy generation methods, training models specific to the number of fragments, engineering of ML features, comparing other ML methods and evaluating other MS/MS similarity metrics such as the spectral entropy[6], our present results show better performance over existing LC-IM-MS tools for confident metabolite annotation with PeakDecoder using ML features based on summary statistics and an SVM classifier. Limitations of the current algorithm include requirements for sufficient high-quality peak-groups for training (i.e., limited performance for samples with very low complexity) and a library acquired with compatible analytical conditions for inference.

Since the training strategy in PeakDecoder is to learn how to distinguish good co-elution and co-mobility patterns from the raw data directly and does not rely on fragmentation rules, its application is not limited to a particular omics. We believe that PeakDecoder represents a step towards universal software for molecular identification and it will potentially enable error rate calculations for different analyte types. Future work will be performed to compare PeakDecoder to DDA analyses and to evaluate it with predicted MS/MS, CCS, and RT metabolomics libraries, as well as applications for proteomics and lipidomics. While PeakDecoder was built on several MS-tools, we envision a fully automated pipeline which is enhanced by replacing with novel artificial intelligence (AI)-based methods the traditional tools that heavily require intervention from human experts. Similarly to other research fields, advanced AI MS-tools may achieve human-level or super-human AI systems[47] and have the potential to exploit the rich multidimensional LC-IM-MS data to derive new molecular knowledge.

## Methods

### Automated chromatographic method selection software
The Automated chromatographic Method Selection Software (AMSS) used PubChem IDs as input for information on molecules such as SMILES and physical and chemical properties, and utilized the previously published BioCompound Machine Learning (BCML) tool[48] to

calculate additional physico-chemical properties. PaDEL descriptor[49] was used to compute molecular descriptors which are used as features for further ML applications. As the training data provided for developing the ML application was limited, the feature selection method Boruta[50] was applied to avoid overfitting the predictive model. The random forest method was applied in sci-kit-learn for ML predictive model following the feature selection. Datasets from previous HPLC analysis of different compounds from the IROA Compound Test Set (SIGMA Chemical) were analyzed in ESI positive and negative modes using four different chromatography methods (HILIC+, HILIC−, RP+, RP−) and used as training: 467 compounds for pH 9.2 and 508 compounds for pH 2.7. The training datasets had scores for performances of all analysis methods for different compounds. Using the strategy mentioned above, four different categorical predictive models were built for each chromatographic analysis method. The predictive model is used to predict the best chromatograph analysis method for testing compounds. Additionally, Local Interpretable Model-Agnostic Explanations (LIME)[51] was applied for model application explanation. LIME scores were also used to draw shapes with color codes to highlight the chemical structural/substructure of the compounds with prediction. This software is used to run all four predictive models on the new test case. The predictions for all the four different models are reported as predictions. It should be noted that there can be multiple suitable methods for HPLC analysis for a single compound.

### Sample preparation
**Standards.** Sixty-four commercially available compounds from the central carbon metabolism (common to all ABF hosts) or metabolites that are part of pathways that had been engineered in the ABF mutant strains were selected. Standards were prepared individually at a concentration of 25 µM using 3:2 acetonitrile: water as solvent. Once analyzed individually, standard mixes containing 10–15 metabolites at the same final concentration and solvent composition were prepared and acquired in the analytical platforms.

***A. pseudoterreus*** **and** ***A. niger*** **strains.** The *A. pseudoterreus* codon optimized β-alanine pathway was detailed in our previous manuscript[24]. The β-alanine pathway was randomly integrated into *A. niger* ATCC 11414. Three transgenic strains (3HP-10, 5, and 9) producing low, medium, and high levels of 3HP were selected for metabolite profiling. In addition, two transgene overexpression constructs were built. *A. niger* aspartate aminotransferase (*aat*, Genebank access: EHA22111.1) cDNA was under the control of *A. niger* translation elongation factor-1a (*tef1*) promoter and its first intron and the transcriptional terminator of *A. niger* phosphoglycerate kinase (*pgk1*), while *A. niger* pyruvate carboxylase (*pyc*, Genebank access: AJ009972.1) cDNA was under the control of *A. niger* multiprotein-bridging factor-1 promoter and the transcriptional terminator of *pgk1*. Both of transgene expression constructs were separately introduced into strain 3HP-9 to generate a series of new transgenic strains: 3HP-9 *aat*-1 to 12 or 3HP-9 *pyc*-1 to 12. Transgenic strains 3HP-9 *aat*−5 and 3HP-9 *pyc*-1 producing higher and highest levels of 3HP were selected for metabolite profiling. The selected strains were grown in 50 ml of the modified Riscaldati B medium[24] in 250 ml PYREX Erlenmeyer flasks. The flasks were incubated at 30 °C while shaking at 200 rpm. The supernatants and biomass were collected at day 4. For each culture, 2 ml of supernatant was filtered through a 0.2 µm syringe filter and 1 ml of biomass was collected via vacuum filtration through 2 layers of EMD Millipore miracloth and washed with 2 ml of phosphate-buffer saline. The biomass was transferred into 1.5 ml microcentrifuge tubes and immediately frozen in liquid nitrogen. Both supernatants and biomass pellets were stored at −80 °C prior to extraction of metabolites.

***P. putida*** **strains.** Detailed explanation about the integration site selection, plasmid design, assembly, and transformation were

 

presented previously[37]. Briefly, *P. putida* KT2440 was used as the wild-type strain. An mKate fluorescent reporter construct was designed, synthesized, and introduced into seven different insertion locations on the *P. putida* KT2440 chromosome by homologous recombination, always in the same orientation. Growth and fluorescence of these seven mKate expression strains were measured in M9 minimal medium containing 30 mM glucose and reported in the cited manuscript. To further test the effect of the integration locus on function of a heterologous gene, a functional protein (muconate transporter) was integrated into three of the seven sites and the resulting variation in growth and protein expression was measured. The selection of the integration sites chosen for additional characterization was based on their display of different phenotypes with mKate, such as slow growth and low fluorescence (PP 2224), slow growth and high fluorescence (PP 1642), or WT growth and medium fluorescence (PP 5042). Overnight cultures of WT *P. putida KT2440* and strains carrying a codon-optimized copy of mucK (the gene coding for the muconate importer) were inoculated into 10 mL LB medium to give a starting culture density of 0.2 OD600 nm and were incubated at 30 °C with shaking until the culture density reached 1.0 OD600 nm. Cell cultures were centrifuged and washed twice in M9 salts before resuspending in the same buffer. The washed cells were used to inoculate 50 mL M9 medium containing 30 mM glucose (for WT KT2440) or 30 mM cis,cis-muconate (for strains containing mucK insertions). The starting culture density was 0.1 OD600 nm and growth continued until 0.7 OD600 nm was reached. Cells were collected by centrifugation and were washed one time with ice-cold PBS. Cell pellets were weighed (~50 mg of wet weight collected), flash frozen in liquid nitrogen, and stored at −80 °C prior to shipment and extraction of metabolites and proteins. Omics samples were prepared in triplicate.

**R. toruloides strains.** The *R. toruloides* strain used in this study, GB2, was described in detail in our previous manuscript[42]. Its parent strain, BIS3, was the highest bisabolene producer of a panel of $P_{GAPDH}$-BIS strains that were modified only by insertion of a heterologous α-bisabolene synthase gene (BIS) from *Abies grandies* under control of the native *R. toruloides* GAPDH (glyceraldehyde 3-phosphate dehydrogenase) promoter into WT *R. toruloides* and differed in copy number only[40,42]. BIS3, was selected for addition of a second expression cassette consisting of BIS under control of the native *R. toruloides* ANT (adenine nucleotide translocase) promoter, which resulted in strain GB2. GB2 contains 6 copies of the $P_{ANT}$-BIS cassette in addition to the original 10 copies of the $P_{GAPDH}$-BIS cassette in BIS3. GB2 cells were grown in vessels in an Ambr® 250 High Throughput system (Sartorius) with a total volume of 150 ml each. The growth media consisted of four DMR (deacetylation and mechanical refining method) hydrolysates made from corn stover by the National Renewable Energy Laboratory (Golden, Colorado) from a 2 × 2 matrix of ash (high/low) and moisture (high/low). The DMR hydrolysates were referred to as HALM (high ash, low moisture), HAHM (high ash, high moisture), LALM (low ash, low moisture), and LAHM (low ash, high moisture). The media were only supplemented with ammonium sulfate (5.00 g/L), potassium phosphate monobasic (10.34 g/L) and potassium phosphate dibasic (4.18 g/L), pH was controlled at 5.0 by addition of ammonium hydroxide. Dissolved oxygen was set as 30%, air flow 75 standard liter per minute (=0.5 volume of air sparged in aerobic cultures per unit volume of growth medium per minute), agitation (cascade) of 500–2000 rpm and growth temperature of 30 °C. A dodecane overlay (20% of total volume) was added to capture the bisabolene produced. Three bioreactors were prepared for each condition (hydrolysate). For omics measurements, 5 mL volume of culture were taken from each Ambr fermentation vessel at time points 24 and 60 h and centrifuged at $4000 \times g$ at 4 °C for 5 min. The supernatant and dodecane overlay were decanted and transferred to another tube for bisabolene analysis, done by GC-MS as described previously[52]. The cell pellet was

resuspended in 1.5 ml of ice-cold PBS and transferred to a new tube. Samples were centrifuged for 5 min at $16,000 \times g$, the PBS removed, and the cell pellet was flash-frozen with liquid nitrogen. Pellets were stored at −80 °C until shipment and extraction of metabolites and proteins.

All microbial samples (cell pellets) were extracted using the MPLEx protocol as previously reported[24,37,53]. Briefly, a mixture of chloroform, methanol, and water was added to the cell pellets, extraction done in an ice bath and the polar and non-polar phases were combined and dried under vacuum. Dried extracts were resuspended in 300 μl of 3:2 acetonitrile:water, transferred to an LC-MS vial, and stored at −20 °C until analysis.

## SRM and LC-IM-MS analyses

Ultrahigh performance liquid chromatography (UHPLC) methods were implemented and optimized by analyzing standards. Chromatographic separation was performed with an Agilent UHPLC 1290 Infinity II system. The sample injection volume was 10 μL and the autosampler temperature was maintained at 4 °C. The Agilent UHPLC was equipped with a Water XBridge BEH Amide XP Column, 2.5 μm (2.1 mm i.d. X 50 mm). A Waters XBridge BEH Amide XP VanGuard cartridge, 2.5 μm (2.1 mm i.d. X 5 mm) was also installed to remove potential particulate contamination from the mobile phases. Mobile phases consisted of (A) 10 mM ammonium acetate, 10 μM InfinityLab deactivator additive, pH 9.2 in 90% water and 10% acetonitrile, (B) 10 mM ammonium acetate, pH 9.2 in 90% acetonitrile. The column was kept at 50 °C throughout the run. The gradient length was 8.70 min (detailed as following, 0.0:0.350:90, 1.0:0.350:90, 1.1:1.0:85, 4.0:0.750:80, 5.0:0.750:40, 6.5:0.750:40, 6.8:0.750:20, 7.0:0.750:20, 7.5:0.750:90 in terms of min:flow-rate-μL/min:%B) with an equilibration time of 3.0 min. The UHPLC system was coupled to an Agilent 6490 triple quadrupole (QQQ) for initial method development. Scan and SRM analyses were performed for precursor fragmentation and transition identification. The instrument was operated in the negative polarity with the following parameters: ion spray voltage of 3000 V, capillary inlet temperature of 225 °C, gas flow 15 ml/min, nebulizer pressure 20 psi, sheath gas temperature 250 °C, sheath gas flow 11 ml/min. Data were acquired in a mass range from 65 to 1400 *m/z*. SRM analyses were also performed for the calibration curves of example standards and the evaluation in microbial samples. Data was processed in Skyline[14] (v.64.21.1.0.146) for peak area integration. A total of 11 samples were analyzed by SRM for the *P. putida* strains, with 3 biological replicates for all, except mucK PP2224 that had 2).

The optimized UHPLC system was coupled to an Agilent 6560 Drift Tube Ion Mobility Spectrometry (DTIMS)-QTOF MS (Agilent Technologies, Santa Clara, CA). The MassHunter data acquisition software (v.B.09.00 (B9044.0), Agilent Technologies) was used to collect all mass spectrometry raw data files. The instrument was mass-calibrated before every batch measurement using the Agilent ESI Tune solution. Standard mixes and microbial samples were analyzed in negative mode using a Dual AJS ESI and high-purity nitrogen as drift gas. Parameters were set to 325 degrees C gas temperature, 5 L/min drying gas, 30 psi nebulizer, 275 degrees C sheath gas, 11 L/min sheath gas flow, 2500 V Vcap, 2000 V nozzle voltage, and 400 V fragmentor. Data was acquired in All-Ions DIA mode alternating between low (MS1) and high (MS2) collision energies at the frequency of 2 frames per second. 60 ms of maximum drift time was allowed with 19 transients per frame. Mass range 50–110 *m/z*. Fixed CE values of 20 or 40 V were used to cover both labile and compounds with masses >600 Da. A total of 81 microbial samples were analyzed by LC-IM-MS: 46 for the *A. pseudoterreus* and *A. niger* strains (4 biological replicates for each condition, except groups Control (Exp 1, A. pseudoterreus cad) and F (Exp. 1. A. niger 3-HP high) that had 3 each; analyzed with 20 V CE only), 11 for the *P. putida* strains (3 biological replicates for all, except mucK PP2224 that had 2; analyzed with 20 and 40 V CEs), and 24 for

the *R. toruloides* strains (4 biological replicates for each group; analyzed with 20 and 40 V CEs).

## Data processing for LC-IM-MS

CCS were calculated using the IM-MS Browser (v.10.0, Agilent Technologies) with the single-field method[54] and the Agilent Tune-Mix solution as calibrants. The PNNL-Preprocessor[55] (v2020.07.24) was used to apply moving average smoothing (points: 3 in LC and 3 in IM) and filtering (minimum intensity threshold 20 counts). MS-DIAL[11] (v.4.70) was used to perform untargeted feature finding and MS/MS deconvolution (soft ionization, ion mobility separation, data independent MS1, MS, and MS/MS profile data, negative ion mode, metabolomics, centroid MS1 tolerance 0.01, centroid MS2 tolerance 0.025, smoothing level 1, minimum peak width 3, minimum peak height 300, peak spotting mass slice width 0.1, deconvolution sigma window 0.5, MS2Dec amplitude cut off 0, alignment RT tolerance 0.1, alignment MS1 tolerance, 0.015, alignment RT factor 0.5 and ion mobility accumulated RT range 0.2). Skyline[14] (v.64.21.1.0.146) was used to perform targeted data extraction (acquisition method DIA, isolation scheme All Ions, mass analyzer TOF, mass resolving power 10,000, ion mobility resolving power 40, and small molecule fragment types "p,f"). Implementation of the PeakDecoder algorithm and evaluation of the results were performed in R (v.4.1.0, A language and environment for statistical computing, R Foundation for Statistical Computing, Vienna, Austria, https://www.R-project.org), using packages e1071 (v.1.7-9) and ggplot2 (v.3.3.3).

The following approximation was used to calculate the negative mobility offset of the fragments from their precursors: $((FragmentMz - PrecursorMz)/PrecursorMz)*0.7 - PrecursorMz*0.0001$. Which overall worked well for both 20 V and 40 V collision energies and resulted in values mostly between −0.1 and −0.3 ms, with smaller *m/z* ions showing larger offsets. This negative drift time shift is a function of the collision energy used and the mass of the fragment ion and it can be explained by the fact that under the accelerating electric field smaller ion fragments move faster through the collision cell and the ion beam compressor region during high-energy steps than larger precursor ions[33]; hence t0, i.e., the time ions spend traveling though the instrument, outside the drift tube, is different.

A library with precursor *m/z*, RT, CCS, and fragment *m/z* values was built from the standards. RT and transitions were obtained from the SRM results. CCS and additional transitions were determined post-acquisition from the LC-IM-MS data. The list of 64 metabolites with accurate mass, RT, and CCS are in Supplementary Table 1. The list of fragments in csv format and the full library in the NIST MSP text format library for metabolite identifications are in Supplementary Data 1.

## Metabolite scoring for LC-IM-MS: PeakDecoder algorithm

(1) Feature finding and fragment ion deconvolution: data is processed in untargeted mode (MS-DIAL) to extract all precursor ion features (MS1) and their respective deconvoluted fragment ions (MS2) based on co-elution and co-mobility. The alignment (Peak ID matrix, msp format) and all peak lists (txt, centroid) are exported.

(2) Target and decoy generation: an R script was implemented to generate a training set. The MS-DIAL alignment results including features and their fragments is used as input. Features with S/N > = 15 and at least 3 fragments with intensities within 1–130% of their precursor intensity are kept as targets. The top 16 most intense fragments are kept per target.

To generate the decoys, the set of targets are associated by pairs. For each target, another target is found from the same representative LC-IM-MS run, which the precursor *m/z* is within 50 units difference (to ensure that the paired features are from molecules of similar size), has the largest RT difference (at least

3 min to avoid pairing a repeated feature from a large tailing peak) and has the same number of fragments. A pair of decoys is generated for the paired targets by keeping the same precursor properties and swapping the *m/z* values of 40–60% of the fragments randomly chosen from the top-most-intense. Targets for which decoys could not be generated are excluded. A transition list in Skyline format is generated with this preliminary set of targets and decoys.

(3) Targeted data extraction: the transition list for the training set and the query metabolites in the library are processed separately. The precursor and fragment ion signals are extracted (Skyline) from all the LC-IM-MS runs. The two reports (training set and query metabolites) are exported, which include the required XIC metrics: area, height, mass error, FWHM (LC), RT, expected RT, expected CCS.

(4) Machine learning training: an R script was implemented for training. The Skyline report of the preliminary training set is used as input. The targets are filtered according to multiple thresholds to ensure a good quality training set. Fragments with unassigned height (i.e., NA or zero) and precursors with S/N < 20 are removed. Each fragment is evaluated to count the number of low-quality metrics: area < = 0, height <1% their precursor height, mass error >15 ppm, RT difference to their precursor larger than 0.1 min, and FWHM difference to their precursor larger than 2x the precursor FWHM. Targets with at least 2 fragments with high-quality metrics are kept. To simulate interferences some fragments with low-quality metrics are kept. Fragments with the worst metrics and ranked higher than 2x the number of fragments with good metrics are removed. For each LC-IM-MS run, only the paired decoys with the same subset of fragments as their targets after filtering are kept for maintaining the same distribution of targets and decoys by number of fragments and *m/z* values. The target fragment height is used as the expected intensity and assigned to the corresponding decoys to minimize the impact of peak integration differences between MS-DIAL and Skyline. The filtered targets with at least 3 fragments in total and their corresponding decoys are used as the final training set. The following descriptors are calculated for the filtered training set and used as ML features:

- DIA-cosSim: cosine similarity between the integrated area and the expected intensity of the fragments.
- DIA-RTdiffSd and DIA-RTdiffMean: standard deviation and mean of the differences between the precursor RT and RT of its fragments.
- DIA-FWHMdiffSd and DIA-FWHMdiffMean: standard deviation and mean of the differences between the precursor FWHM and FWHM of its fragments.
- DIA-MassErrorSd and DIA-MassErrorMean: standard deviation and mean of the fragment mass errors.

An SVM binary classifier (e1071 R package) is trained using a radial kernel, scaling (to zero mean and unit variance), 10-fold cross validation, and probability calculation. The probability is calculated by fitting a logistic distribution using maximum likelihood to the decision values of all binary classifiers, and computing the a-posteriori class probabilities for the multi-class problem using quadratic optimization. The trained model is saved. The target probabilities are calculated for the full training set and a confusion matrix and FDR are calculated to evaluate performance. The FDR is calculated as $FP/(TP + FP)$[56]. The target probability is used as the PeakDecoder score. A table with pairs of values (FDR, PeakDecoder score) is automatically generated after training (file PeakDecoder-FDR-thresholds_[dataset].csv).

(5) Machine learning inference: an R script was implemented for inference. The model previously trained and saved is loaded. The Skyline report for the query metabolites and the library with the

fragment ion abundances generated from the standards are used as input. The descriptors are calculated as described above. The precursor RT error (minutes) is calculated as the difference between the run RT and the expected RT (from the standards). The CCS error is calculated as the percentage difference between the run CCS (obtained from the corresponding MS-DIAL peak lists, since Skyline uses the CCS as a filter and does not report the actual CCS from the IM peak apex in the run) and the expected CCS (from the standards). A query metabolite is considered identified if, in at least one of the runs, passes all cutoff thresholds: precursor mass error <18 ppm, precursor RT error <0.4 min, CCS error <0.8%, and PeakDecoder score >0.8 (or corresponding to 1% FDR).

## Pathway analyses

Metabolites with at least 1 replicate identified with high confidence were selected and their Skyline-integrated precursor and fragment ion abundances across all runs were used for quantitation. Statistical analysis of the metabolite abundance data was performed in R using the pmartR package[57] (v0.9.0). For *P. putida* and *R. toruloides* data-sets, the mean values of abundances acquired with 20 V and 40 V CE were used for the analysis. The abundance values were log2 trans-formed, and the test for differential abundance between control and test samples was performed using the IMD-ANOVA method[58]. Clus-tered heatmaps of log2 abundances were generated using the R package pheatmap (v1.0.12) with Euclidean distance and complete linkage method. Bar and error bar plots shown on the metabolic pathway maps were generated using the python package matplotlib (v3.5.1). The metabolic pathway maps for *A. pseudoterreus/A. niger* and *R. toruloides* were drawn based on the genome-scale metabolic models iJB1325[59] and Rt_IFO0880[60] using ChemDraw (v19.0). The metabolomics and proteomics data visualization on the *P. putida* metabolic pathway map was performed with the genome-scale metabolic model iJN1462[61] using the python packages escher[62] (v1.7.3) and cobrapy[63] (v0.22.1).

## GC-MS analyses

Dried extracts for metabolomics analysis were obtained after MPLEx extraction as explained in the Sample Preparation section. The stored metabolite extracts were completely dried under speed-vacuum to remove moisture and chemically derivatized as previously reported[64]. Briefly, the extracted metabolites were derivatized by methox-yamination and trimethylsilylation (TMS), then the samples were ana-lyzed by GC-MS. Samples were run in an Agilent GC 7890A using a HP-5MS column (30 m × 0.25 mm × 0.25 μm; Agilent Technologies, Santa Clara, CA) coupled with a single quadrupole MSD 5975C (Agilent Technologies). One microliter of sample was injected into a splitless port at constant temperature of 250 °C. The GC temperature gradient started at 60 °C and hold for 1 min after injection, followed by increase to 325 °C at a rate of 10 °C/min and a 5-min hold at this temperature. Fatty acid methyl ester standard mix (C8-28) (Sigma-Aldrich) was analyzed in parallel as standard for retention time calibration. GC-MS raw data files were processed using the Metabolite Detector (v2.5)[65]. Retention indices (RI) of detected metabolites were calculated based on the analysis of a FAMEs mixture, followed by their chromatographic alignment across all analyses after deconvolution. Metabolites were initially identified by matching experimental spectra to a PNNL aug-mented version of Agilent GC-MS metabolomics Library, containing spectra and validated retention indices for over 850 metabolites. Then, the unknown peaks were additionally matched with the NIST17/Wiley11 GC-MS library. All metabolite identifications and quantification ions were validated and confirmed to reduce deconvolution errors during automated data-processing and to eliminate false identifications. A total of 46 samples of the *A. pseudoterreus* and *A. niger* strains were analyzed by GC-MS, with 4 biological replicates for each condition,

except groups Control (Exp 1, A. pseudoterreus cad) and F (Exp. 1. A. niger 3HP high) that had 3 each.

## Targeted proteomics analyses of *P. putida* strains

Intracellular proteins from samples of *P. putida* strains, KT2440, with heterologous gene insertion were extracted and digested as previously described[37]. Peptides from previously established assays[39] were used for the targeted proteomics analysis of enzymes in various metabolic pathways. Analysis of the targeted proteomics assay was performed via LC-SRM. To facilitate protein quantification, crude heavy peptide mixture stock solution was spiked in the 0.20 μg/μL peptide samples at a nominal concentration of 25 fmol/μL for each peptide. LC-SRM analysis utilized a nanoACQUITY UPLC® system (Waters Corporation, Milford, MA) coupled online to a TSQ Altis triple quadrupole mass spectrometer (Thermo Fisher Scientific). The UPLC® system was equipped with an ACQUITY UPLC BEH 1.7 μm C18 column (100 μm i.d. × 10 cm) and the mobile phases were (A) 0.1% formic acid in water and (B) 0.1% formic acid in acetonitrile. 2 μL of sample (i.e., 0.4 μg of peptides) were loaded onto the column and separated using a 110-min gradient profile as follows (min:flow-rate-μL/min:%B): 0:0.4:1, 6:0.6:1, 7:0.4:1, 9:0.4:6, 40:0.4:13, 70:0.4:22, 80:0.4:40, 85:0.4:95, 91:0.5:95, 92:0.5:95, 93:0.5:50, 94:0.5:95, 95:0.6:1, 98:0.4:1. The LC column was operated at a temperature of 42 °C. The TSQ Altis triple quadrupole mass spectrometer was operated with ion spray voltages of 2100 ± 100 V and a capillary inlet temperature of 350 °C. Tube lens voltages were obtained from automatic tuning and calibration without further optimization. Both Q1 and Q3 were set at unit resolution of 0.7 FWHM and Q2 gas pressure was optimized at 1.5 mTorr. The transitions were scanned with a dwell time of 0.78 ms. Targeted proteomics data were imported into Skyline (v64.22.2.1.278)[66] and the peak boundaries were manually inspected to ensure correct peak assignment and peak boundaries. Peak detection and integration were determined based on two criteria: (1) the same LC retention time and (2) approximately the same relative peak intensity ratios across multiple transitions between the light peptides and heavy peptide standards. The total peak area ratios of endogenous light peptides and their corresponding heavy isotope-labeled internal standards from Skyline were used for sub-sequent protein abundance rollup and pathway analysis.

## Global proteomics analyses of *R. toruloides* strains

Intracellular proteins from samples of bisabolene producing *R. tor-uloides* strains, GB2.0, were extracted, digested with trypsin, and analyzed by LC-MS/MS following a previously established protocol[60]. A Q-Exactive Plus Orbitrap mass spectrometer (Thermo Fisher Scientific) was used in this study with the parameters as following: full MS (AGC, $3 \times 10^6$; resolution, 70,000; *m/z* range, 300–1800; maximum ion time, 20 ms); MS/MS (AGC, $1 \times 10^5$; resolution, 17,500; *m/z* range, 200–2000; maximum ion time, 50 ms; minimum signal threshold, $5 \times 10^3$; isolation width, 1.5 Da; dynamic exclusion time setting, 30 s; collision energy, NCE 30; TopN, 12). The MS data were searched against the R. tor-uloides strain IFO0880 (v4.0) and heterologous protein sequences [https://mycocosm.jgi.doe.gov/Rhoto_IFO0880_4][45] using MaxQuant[67] (v1.6.2.10) and the following parameters: 1% peptide-level and protein-level FDR, match-between-runs enabled, partial tryptic with trypsin/P, maximum missed cleavage of 2, dynamic modification of oxidation on methionine and N-terminal acetylation, fixed carbamidomethyl on cysteine, mass tolerances of 20 ppm for both precursor and fragment ions, minimum peptide length of 7, and a minimum number of unique peptides for protein quantification as 1. Peptide intensity level data from MaxQuant were further processed by pmartR (v0.9.0) for quality control, protein rollup, and statistical comparisons.

## Reporting summary

Further information on research design is available in the Nature Portfolio Reporting Summary linked to this article.

## Data availability

The microbial LC-IM-MS data (and related Skyline projects) generated in this study have been deposited in the MassIVE database under accession code MSV000089733. The *P. putida* targeted proteomics data generated in this study have been deposited in the Panorama database [https://doi.org/10.6069/6j7y-t592]. The *R. toruloides* global proteomics data generated in this study have been deposited in the MassIVE database under accession code MSV000091202. Source data are provided with this paper.

## Code availability

The source code of PeakDecoder[68], the library built from standards, and all the input files and results can be found at https://github.com/EMSL-Computing/PeakDecoder. The source code of the automated chromatographic method selection software can be found at https://github.com/poorey/AMSS.

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

## Acknowledgements

This work was part of the DOE Agile BioFoundry (http://agilebiofoundry.org) supported by the U.S. Department of Energy, Energy Efficiency and Renewable Energy, Bioenergy Technologies Office (to K.E.B.J.), and used capabilities developed under the National Institute of General Medical Sciences grant P41 GM103493 (to R.D.S.). Microbial biomass samples from fermentations on different hydrolysates were generated as a part of the Feedstock Conversion Interface Consortium, funded by the Department of Energy Bioenergy Technologies Office.

## Author contributions

K.E.B.J., A.B., and N.M. planned this work. A.B. conceived the PeakDecoder algorithm, implemented the software, performed the LC-IM-MS data analyses, and wrote the manuscript. N.M. prepared the metabolomics standards and microbial samples for LC-MS acquisition, optimized the LC method, performed the SRM and GC-MS metabolomics experiments and wrote the interpretation of the *P. putida* and *R. toruloides* results. J.K. performed all the metabolite clustering, pathway and statistical analyses and wrote the interpretation of the *A. pseudoterreus* and *A. niger* results. D.J.O. optimized the method parameters and acquired the LC-IM-MS data. Y.G. performed the SRM and global proteomics experiments and data analysis. K. Poorey and A.G. implemented the AMSS tool. K.W. provided support for optimizing the LC method and

contributed to the development of the SRM metabolomics method. M.B. and C.D.N. performed sample extraction and preparation. K. Pomraning, S.D., and Z.D. planned the experimental design, engineered, and grew the *A. pseudoterreus* and *A. niger* strains. R.W. planned the experimental design, engineered, and grew the *P. putida* strains. E.O. planned the experimental design, engineered, and grew the *R. toruloides* strains. E.S.B. provided insightful comments about IM. C.J.P., R.A.F., A.T., and A.A. contributed to the development and optimization of the LC method and the AMSS tool. Y.K. supervised the initial LC method development and provided helpful comments about metabolomics. D.T., J.G., R.D.S., J.K. Michener, J.M.G., and J.K. Magnuson managed related projects. K.E.B.J. supervised this work and assisted in preparing the manuscript. All authors discussed, edited, and approved the final manuscript.

## Competing interests

A.G., R.A.F., A.T., and A.A. are employees at Agilent Technologies. The remaining authors declare no competing interests.
