## [Peer Review File · Nature Communications]

REVIEWER COMMENTS

Reviewer #1 (Remarks to the Author):

Comments for authors:

In this manuscript, Bilbao et al. developed a PeakDecoder workflow to evaluate false discovery rates in LC-IM-MS DIA data. The authors then applied this approach in three different biological cases of engineered strains. The FDR estimation is an important topic in metabolomics, and the reviewer appreciate authors' attempts in LC-IM-MS data. Nevertheless, similar ideas have been reported, e.g. target decoy (10.1038/s41467-017-01318-5; 10.1021/acs.analchem.9b03355), decoy spectra extraction in DIA data (DIAMetAnalyzer: 10.1038/s41467-022-29006-z). Therefore, a more comprehensive summary is needed to highlight the differences of this work. In addition, more evaluations and comparisons of FDR approach should be added, especially in aspect of FDR reliability. Finally, as a reviewer and reader, I think the listed 3 biological applications seem to be irrelevant to demonstrating strengths of their approach. I hope my suggestions can help authors to improve this work.

1. In introduction, these existed approaches need to be mentioned and summarized.

☐ DIA data processing: MetDIA (10.1021/acs.analchem.6b02122), MetaboDIA (10.1021/acs.analchem.6b05006), DaDIA (10.1021/acs.analchem.0c05022), DecoID (10.1038/s41592-021-01195-3).

☐ FDR estimation in metabolomics: pySM (10.1038/nmeth.4072), Passatutto (10.1021/acs.analchem.9b03355), JUMPm (10.1021/acs.jproteome.8b00019)

2. The first step of this workflow is target generation. The targets for training are mainly relied on MS-DIAL outputs. As one can expect, the performances of machine learning model may be directly affected by MS-DIAL. Do different parameters of MSDIAL affect PeakDecoder performance? How about other deconvolution tools? In addition, different biology samples have different complexities. May it affect the FDR estimation? I think the addition of these evaluations will enhance the robustness of this tool.

3. Then is targeted decoy. I found the decoy spectrum had 3 criteria according to the Method: m/z error difference ≤ 50 Da, RT difference ≥ 3 min (select spectrum the largest error), and same number of fragments. Then, swap partial fragments. My question is why the authors select the decoy spectrum with same fragment number? It may remove a lot of spectra. How about author pooling fragments for spectra with first two criteria, and then randomly replace partial fragments?

4. Model training. Here, the author used 7 features to training the model. I have some concerns about them. Take "DIA-RTdiffMean" as an example. In decoy spectrum generation, one of most important criteria is RT difference ≥ 3 min. This means that most fragments in decoy spectra should have this value larger than 3 min against RT of target. If so, these decoy features can be distinguished easily only using this feature. Whether it cause the underestimate the FDR? Can authors evaluate the reliability of FDR estimation (i.e. estimated FDR vs. actual FDR) using a ground truth data? I think the Passatutto and DIAMetAlyzer are good examples for FDR reliability evaluation.

5. Output. How can users link the PeakDecoder score with FDR? How to select proper thresholds of PeakDecoder score. Common user would prefer directly selecting the FDR level instead of PeakDecoder score.

6. The Figure 2b. As I mentioned in comment 4, the authors need to demonstrate the reliability of FDR. The different distributions of SVM features between target and decoy indicate that this decoy strategy may not provide an accurate FDR estimation. Therefore, a proper evaluation of FDR reliability is required. In addition, please provide distributions of other features, like DIA-RTdiffMean, DIA-FWHMdiffMean, and DIA-MassErrorMean.

7. I don't understand the statistics in table 1. What are definitions of "#unique identified metabolites" and "#Annotated metabolite features"? How did authors annotate these metabolites and features with a small library (64 metabolites)? Why "RT-CCS-DIA" have more annotated metabolites than "RT-CCS" when it has additional constrains?

8. Biology applications. I think listed 3 biological applications in manuscript are irrelevant to demonstrating strengths of their approach. In my opinion, these results seem to be easily done directly using Skyline. I suggest the authors comparing differences of annotated metabolites and conclusions, e.g. using or not using PeakDecoder, or different FDR levels. Such comparisons would be helpful to show benefits of PeakDecoder for biological applications.

9. Code Availability. Source code, demo code, example data, and detailed documents are required. It will help users to use this tool and reproduce results.

10. Supplementary Figure S2: I'm curious about the linear range. Why the LC-IM-MS data have wider linear range than the SRM?

Minors:

11. Some typos:

☒ Legend of figure 1. "...the query set of metabolites and estimate and FDR." Remove second "and".

☒ "Pseudomonas putida is a very promising industrial host,, with biochemical properties that make it ideal for hosting biochemical transformations". Remove the replicated comma.

Reviewer #2 (Remarks to the Author):

In "PeakDecoder: A Machine Learning-Based Metabolite Identification Algorithm for Multidimensional Mass Spectrometry Measurements and its Application in Synthetic Biology" the authors describe a machine learning-based feature detection and deconvolution tool for DIA LC-IM-MS data including FDR estimation and a nice demonstration of their tool for strain optimization experiments with different bacteria.

While I think that the work and novel tools for metabolite annotation are very important and needed for metabolomics DIA and ion mobility data, there are a couple of important points I would suggest the authors should carefully consider and revise, perhaps with additional experiments, before the paper can be published.

While I am not an expert in machine learning, I have some doubts about the training data size and the generation thereof.

In particular, the use of another DIA deconvolution tool, but no ground truth, to generate the training data makes me wonder whether the tool is at risk of over-fitting and if it can perform better /be more sensitive than the initial tool.

Perhaps I have overlooked a critical discussion / comparison between the two. If not done yet, I would suggest doing this. Also a cross validation with other data sets, including ground truth (internal standards) that were not part of the training data might be useful.

I would also suggest discussing more critically the alternative use of DDA as a still widely used and established MS/MS strategy in metabolomics. Here it would have been nice to also compare the two. How many more metabolites can you ID by your DIA method and your in-house library in comparison to DDA and other public MS/MS libraries? Perhaps this could also serve to orthogonality confirm Metabolite IDs from PeakDecoder.

For the FDR estimation, this is great that the authors added this to their tool. Robust ways to estimate FDR in non-targeted metabolomics are certainly needed. While I like the simple concept of the precursor mass swapping method, I wonder how well this performs, also in comparison to other approaches.

Did you do any comparison here? Are there experience/infos available from bench-marking FDR metrics for PSM? I would suggest dedicating more attention to this part and critically discuss/evaluate this.

Detailed Comments:

Formatting:

I would suggest including line numbers for submissions of future/revised manuscripts.

Title:

Maybe better use "Metabolite Detection Algorithm" or "Metabolite Annotation Algorithm" as I assume your tool does not necessary provide Level 1 IDs for compounds detected.

Abstract:

You indicate that PeakDecoder™ is a Trademark. In the conflict of interest statement you further indicate that one of the Authors is an employee of Agilent. Perhaps it would be good to expand here and clarify how much Agilent is involved in software development and IP and whether there could be any conflict of interest, particularly in bench-marking the tool.

Page 2 3rd paragraph:

According to your statement of "hundreds to thousands of primary metabolites and a multitude of secondary metabolites" one might assume that the chemical diversity is greater in primary metabolism, which I assume the authors did not want to claim. Perhaps, specify this more in detail also whether this is with regards to a single organism, or all metabolites in nature.

Page 3 paragraph 1:

I would suggest to point put more clearly the current limitation of DIA for metabolomics. Perhaps it would be good to mention studies that compare DIA to DDA.

Further, while I agree that there is no generally accepted method for FDR estimation in metabolomics, there have been several methods proposed. Perhaps it would be good to mention some of them and elaborate on the disadvantages/challenges here.

Page 7 last paragraph:

What is the number of standards/ library size?

Page 8:

In how many cases could you differentiate co-eluting metabolites. Did you compare the overall numbers of metabolite IDs to DDA experiments?

Page 9, 2nd paragraph.

“A total of 2350 metabolite features could be confidently annotated.” Do you mean annotated as metabolites or m/z – RT “Features”? If you mean features, perhaps “detected” would be the appropriate term here and should be changed throughout the manuscript.

If this refers truly to annotated metabolite IDs, this would be quiet amazing. I would suggest to provide the table of those compounds inc. score, mass error etc. in the SI akin to SI Table 1, where so far only 66 metabolites are listed.

Figure 1 and Page 9, 2nd paragraph:

More details need to be given here. How were the “best quality detected” defined, what is the typical size of training data and perhaps visualize in the figure how the decoys are generated

Table 1:

See my comment above regarding the term “annotated”. Also did you compare the annotation / Identification rate to more standard DDA approaches? From my own experience I would assume that typically more than 28 metabolites should be IDed when using high-quality data and contemporary spectral libraries (Massbank, NIST2020, Metlin, GNPS etc.).

Did you match your deconvoluted DIA spectra against any public MS/MS libraries, or only against the in house library. Perhaps other DIA databases such as “Weizmass” (Shahaf et al. Nat Com) might also be relevant here.

Page 13:

How was the quant experiments for metabolites / proteins performed. Internal /external standards?

Do you have absolute concentrations? (Like for Bisabolene) Or do you refer to relative quant?
Perhaps state this in the text / figure title and caption.

Figure 4:

The quantification refers to relative quant or absolute quant with authentic standards.

Also why is the here presented set of compounds smaller than the list in the SI or the above mentioned 2350 metabolite features.

Page 16, Discussion:

Why will LC-IM-MS and PeakEncoder enable faster testing of strains? I would argue that conventional DDA LC-MS/MS and other state of the art feature finding tools will result in the same speed. On the other hand, MS1 based approaches such a FIA or Rapidfire are likely significant faster. Before overstating speed here, the advantages should eb clearly discussed and perhaps typical analysis/processing times should be given.

Page 17 2nd paragraph:

Awesome that you made the library available to the community. Perhaps infos about file format and a download link would be nice to have here too.

Daniel Petras

Reviewer #3 (Remarks to the Author):

The authors propose a combination of analytical and computational methods for metabolite profiling based on LC-IMS-MS in DIA mode. A main contribution is the so-called PeakDecoder-

algorithm to score library hits. The algorithm consists of several stages, some of them based on support vector machines.

Particularly interesting is the strategy to generate decoys, which seems logical and probably generalizable to other fields apart from metabolomics that should be just as appropriate as the strategy layed out in reference 20 of the proposed manuscript.

Considering how important accurate identification of metabolites is for many different application scenarios, I would expect the work to have significant impact in the field.

The methodology is sound and the conclusions are supported by the analysis and the presented data. The computational building blocks are comparably simple. While it makes sense, in my opinion, to err on the side of simplicity rather than complexity, a sensitivity analysis for some of the choices (mostly cosine distance for comparison and SVMs as classifier) would have been nice, but I strongly suspect that it would not change that much.

In summary, I consider the proposed workflow as a very interesting alternative to the state of the art with the potential of significant impact, and would recommend to accept the manuscript for publication.

Response to reviewers

Manuscript ID: NCOMMS-22-26518-T

Title: "PeakDecoder enables machine learning-based metabolite identification and accurate profiling in multidimensional mass spectrometry measurements"

Authors: Aivett Bilbao, Nathalie Munoz, Joonhoon Kim, Daniel J Orton, Yuqian Gao, Kunal Poorey, Kyle R. Pomraning, Karl Weitz, Meagan Burnet, Carrie D. Nicora, Rosemarie Wilton, Shuang Deng, Ziyu Dai, Ethan Oksen, Deepti Tanjore, James Gardner, Richard D. Smith, Joshua K. Michener, John M. Gladden, Erin S. Baker, Christopher J. Petzold, Young-Mo Kim, Alex Apffel, Jon K. Magnuson and Kristin E. Burnum-Johnson

Authors' response: We sincerely thank the three reviewers for taking the time to review our manuscript, and we very much appreciate their insightful comments; their feedback has allowed us to substantially improve the presentation and robustness of our PeakDecoder workflow. We have carefully revised our manuscript by addressing each of their concerns and recommendations as described in the following sections (authors' response in blue text).

Note: following the Nature Communications formatting guidelines, the title was changed to "PeakDecoder enables machine learning-based metabolite identification and accurate profiling in multidimensional mass spectrometry measurements".

Reviewer #1 (Remarks to the Author):

Comments for authors:

In this manuscript, Bilbao et al. developed a PeakDecoder workflow to evaluate false discovery rates in LC-IM-MS DIA data. The authors then applied this approach in three different biological cases of engineered strains. The FDR estimation is an important topic in metabolomics, and the reviewer appreciate authors' attempts in LC-IM-MS data. Nevertheless, similar ideas have been reported, e.g. target decoy (10.1038/s41467-017-01318-5; 10.1021/acs.analchem.9b03355), decoy spectra extraction in DIA data (DIAMetAnalyzer: 10.1038/s41467-022-29006-z). Therefore, a more comprehensive summary is needed to highlight the differences of this work. In addition, more evaluations and comparisons of FDR approach should be added, especially in aspect of FDR reliability. Finally, as a reviewer and reader, I think the listed 3 biological applications seem to be irrelevant to demonstrating strengths of their approach. I hope my suggestions can help authors to improve this work.

1. In introduction, these existed approaches need to be mentioned and summarized.

DIA data processing: MetDIA (10.1021/acs.analchem.6b02122), MetaboDIA (10.1021/acs.analchem.6b05006), DaDIA (10.1021/acs.analchem.0c05022), DecoID (10.1038/s41592-021-01195-3).

FDR estimation in metabolomics: pySM (10.1038/nmeth.4072), Passatutto (10.1021/acs.analchem.9b03355), JUMPm (10.1021/acs.jproteome.8b00019)

We have added the references for MetDIA, MetaboDIA, DaDIA, DecoID, pySM and JUMPm.

The references for XY-Meta (10.1021/acs.analchem.9b03355) and Passatutto (10.1038/s41467-017-01318-5) were already included in the first version of our manuscript. However, both of those references were placed in the discussion about FDR which was in the Results, and we recognize that this is not the common practice. To improve the clarity of the manuscript, we have introduced other FDR related works with their references in the Introduction and included the additional references recommended for both DIA data processing and FDR estimation. Please see the two paragraphs in the updated Introduction, page 3, line 13 and line 29.

2. The first step of this workflow is target generation. The targets for training are mainly relied on MS-DIAL outputs. As one can expect, the performances of machine learning model may be directly affected by MS-DIAL. Do different parameters of MSDIAL affect PeakDecoder performance? How about other deconvolution tools? In addition, different biology samples have different complexities. May it affect the FDR estimation? I think the addition of these evaluations will enhance the robustness of this tool.

Yes, the performance of the machine learning model can be affected by the deconvolution results. But more than the deconvolution parameters, what will affect the performance of PeakDecoder is the size of the training set. MS-DIAL is so far the only free tool for metabolomics that supports untargeted deconvolution with both LC and IM separations. Besides intensity thresholds, the main deconvolution parameter in MS-DIAL is the sigma window: a higher value (0.7-1.0) will decrease the number of resolved peaks and a lower value (0.1-0.3) may result in many noisy chromatographic peaks. In our results a sigma medium value of 0.5 provided a good compromise.

Importantly, PeakDecoder does not use the MS-DIAL results directly to train the model, it uses that information to generate a preliminary training set as coordinates and performs targeted data extraction with Skyline for both targets and decoys, then it uses the XIC metrics to apply filtering for high-quality fragments to keep high-quality peak-groups as targets and their corresponding decoys. By re-extracting the signals and using only high-quality peak-groups, the potential effect of poorly deconvoluted peak-groups is minimized. Nevertheless, the performance will be affected if the number of targets and decoys are too low or if the classifier results in a close-to-perfect accuracy (>99), since the minimum non-zero FDR that could be estimated is affected due to the small number of false positives. To clarify this point, we added the new Supplementary Fig. 5, improved Fig. 1-b to indicate that there is a preliminary training set and indicated in the legend the filtering of high-quality fragments for the final training set. Furthermore, the following paragraph was added in the section “Developing the PeakDecoder algorithm”:

Since PeakDecoder generates the decoys from unannotated LC-IM-MS DIA experimental spectra, the size of the target library does not affect its performance. However, the performance of PeakDecoder depends on the training set and the validity of the estimated FDR depends on the number of generated false positives. The size and quality of the training set can be controlled in two ways: the parameters of the UFD tool used to generate the preliminary training set (**Fig. 1-b**, Step-1) and the filtering for high-

quality fragments used to generate the final training set (Fig. 1-b, Step-4). At the same time, a tradeoff in the quality of peak-groups is necessary to avoid overfitting and perfect training accuracy, and thus, to estimate a reliable FDR. These components allow the user to define the quality of the resulting annotations and are evaluated using microbial data in the next section.

Different biological samples certainly have different complexities, and this is one of the reasons why we would like to include the 3 synthetic biology applications. We have replaced the term “feature complexity” by “sample complexity” in the discussion of this point and improved the section “Applying PeakDecoder in microbial samples”.

3. Then is targeted decoy. I found the decoy spectrum had 3 criteria according to the Method: mz error difference ≤ 50 Da, RT difference ≥ 3 min (select spectrum the largest error), and same number of fragments. Then, swap partial fragments. My question is why the authors select the decoy spectrum with same fragment number? It may remove a lot of spectra. How about author pooling fragments for spectra with first two criteria, and then randomly replace partial fragments?

We decided to pair features with the same number of fragments as an approach to increase the chances that the molecules are similar and to ensure that the overall distributions of general properties of targets and decoys are the same (please see Supp. Fig. 3). If the number of fragments is not the same for each pair, it becomes more difficult to track and ensure that the distribution of precursors by number of fragments are the same for targets and decoys after applying the filtering for high-quality fragments. Furthermore, pooling fragments and randomly selecting them provides poor performance because it increases the probability to generate unrealistic decoys. This was already found by Scheubert et al. To clarify this point, we have added the following sentence:

Pairing precursors with the same number of fragments was used as an approach to increase the chances that the molecules are similar and to ensure that the overall distributions of general properties of targets and decoys are the same. Generating decoys by pooling and randomly adding fragments was avoided because it has previously shown poor performance (naïve method)²⁰, as it increases the probability of generating unrealistic decoys.

4. Model training. Here, the author used 7 features to training the model. I have some concerns about them. Take “DIA-RTdiffMean” as an example. In decoy spectrum generation, one of most important criteria is RT difference ≥ 3 min. This means that most fragments in decoy spectra should have this value larger than 3 min against RT of target. If so, these decoy features can be distinguished easily only using this feature. Whether it cause the underestimate the FDR? Can authors evaluate the reliability of FDR estimation (i.e. estimated FDR vs. actual FDR) using a ground truth data? I think the Passatutto and DIAMetAlyzer are good examples for FDR reliability evaluation. While the swapped fragments in the decoys have a RT difference ≥ 3 min from their original RT, after decoys are generated, the signals of both targets and decoys are re-

extracted using Skyline (the new Fig. 1-c with the target-decoy example was improved to better explain this concept). The new extracted decoy signal would not result in a RT as large as the original because it is too far from the new decoy precursor RT, and since DIA fragments all ions there is a chance that a signal may be detected at the new decoy RT. This is reflected in the distributions of the targets and decoys by the different individual scores, where the targets and decoys partially overlap, since they cannot be fully discriminated by any of the individual scores alone.

We agree that an evaluation of the reliability of our FDR estimation method was very important and needed, and we appreciate that the reviewer indicated this issue and provided examples. We initially create a list of unique metabolites expected in each dataset but only inspected manually the fragment scores for the most intense replicates. To address this point, we performed a comparison against the ground truth generated from a full dataset manually curated. The results are in the new Fig. 2-c and indicated that the estimated 1% FDR corresponded to a ~2% actual FDR, and while there is an underestimation, it decreased the number of false positives. We recognize that more evaluations are needed with a larger library for an exhaustive validation, but we believe that our results with our current library built from pure standards are sufficient to introduce and demonstrate the potential of our PeakDecoder workflow.

5. Output. How can users link the PeakDecoder score with FDR? How to select proper thresholds of PeakDecoder score. Common user would prefer directly selecting the FDR level instead of PeakDecoder score.

Yes, a table with pairs of values (FDR, PeakDecoder score) is automatically generated after training (file PeakDecoder-FDR-thresholds_[dataset].csv), and this is what the user can utilize to select the corresponding PeakDecoder score threshold. This is now indicated in the Methods.

6. The Figure 2b. As I mentioned in comment 4, the authors need to demonstrate the reliability of FDR. The different distributions of SVM features between target and decoy indicate that this decoy strategy may not provide an accurate FDR estimation.

Therefore, a proper evaluation of FDR reliability is required. In addition, please provide distributions of other features, like DIA-RTdiffMean, DIA-FWHMdiffMean, and DIA-MassErrorMean.

In contrast to classical MS/MS matching in DDA that uses a single or few scores in the mass dimension only, DIA chromatograms from precursor and fragments can provide additional peak shape similarity scores. The distributions in Fig. 2 show the individual scores that are used as features for the SVM, not to estimate the FDR. Each score by itself does not provide a good discrimination between targets and decoys, and therefore they cannot provide an accurate FDR estimation. But when they are combined by the SVM, the final composite score (the PeakDecoder score) provides a great discrimination power between targets and decoys, therefore it can be used to estimate a more accurate FDR. These kinds of overlapping distributions were also found for the individual scores used in the original publication of the mProphet algorithm (Reiter et al. 2011, DOI 10.1038/nmeth.1584, Fig. 2-c shown below), which is the basis for DIA

scoring and FDR estimation with targeted data extraction (i.e., OpenSWATH, pyProphet, DIAMetAnalyzer). The mProphet publication introduced the concept of using decoy transitions at the measurement level for SRM, and this concept was later adapted at the data extraction level for DIA.

To avoid a crowded figure, we initially decided not to include all the distributions of the individual scores in the main manuscript and provided them in the supplementary material. As requested, we have now included all the distributions of the individual scores in Fig. 2., and to clarify how those are used we indicated with labels which distributions represent input as SVM features and which one is the output or composite PeakDecoder score.

7. I don't understand the statistics in table 1. What are definitions of "#unique identified metabolites" and "#Annotated metabolite features"? How did authors annotate these metabolites and features with a small library (64 metabolites)? Why "RT-CCS-DIA" have more annotated metabolites than "RT-CCS" when it has additional constrains?

We are trying to distinguish repetitions due to sample replicates. To simplify the table headers, we are now using the terms "# unique metabolites" and "# annotated features". Unique metabolites do not count the same metabolite identified in multiple samples and conditions. Annotated features count repetitions, i.e., all the features that were annotated, including replicates across multiple conditions. We were able to annotate more features than the number of metabolites in the library because we analyzed many samples across multiple different conditions. We also added the text to the table description: "annotated features include sample replicates and different conditions".

Furthermore, RT-CCS-DIA and RT-CCS do not refer to constrains, these columns in the table are intended to reflect the number of metabolites that were detected including fragments (DIA) or not. While many metabolites could be detected with fragments, some metabolites, particularly the smaller ones, could not be detected with fragments, therefore, the identification is only based on RT and CCS. To clarify this point, we added the text to the table description:

Number of unique metabolites and number of annotated features refer to the identifications obtained by matching against our library of 64 metabolites by accurate mass plus either 2 dimensions with RT-CSS (annotations at the MS1 level only, i.e., no detected fragments) or 3 dimensions with RT-CCS-DIA (i.e., including fragments).

We also note that the numbers in Table 1 were updated after we removed from the library some fragments with zero intensity for 20V CE for standards relevant to the *P. putida* and *R. toruloides* samples.

8. Biology applications. I think listed 3 biological applications in manuscript are irrelevant to demonstrating strengths of their approach. In my opinion, these results seem to be easily done directly using Skyline. I suggest the authors comparing differences of annotated metabolites and conclusions, e.g. using or not using PeakDecoder, or different FDR levels. Such comparisons would be helpful to show benefits of PeakDecoder for biological applications.

We agree that comparing differences in annotated metabolites is important, but we also believe that biological applications are as important. Please note that PeakDecoder is not only about an FDR method. PeakDecoder was developed in the context of a synthetic biology project, and we believe that showing its usefulness for large-scale real applications will increase the adoption, in general, of multidimensional LC-IM-MS analysis for more accurate metabolite annotation and profiling. We added the new Fig. 3 to compare the annotation selectivity by the different analytical separations (mass, mass-RT, mass-CCS and mass-RT-CCS) in microbial samples. We compared metabolites detected in common with GC and SRM platforms in biological samples from the different strains. In the *P. putida* samples, the LC-IM-MS results showed good agreement with corresponding proteomics results. In the *A. pseudoterreus* and *A. niger* samples, by using CCS we could distinguish 3HP from its isomeric molecule lactic in biological samples. In the *R. toruloides* samples, we were able to relate proteomics as well as bisabolene production measured in a dodecane overlay to intracellular metabolites that are intermediates in the mevalonate pathway. The confident identification and accurate quantitation in these studies would not have been possible if we were not using this combination of analytical and computational approach. Using Skyline alone would have not provided the same results, as it does not have a method to calculate a composite score like PeakDecoder nor to estimate FDR for metabolites, and it would have provided more false positives. Furthermore, Skyline does not provide an experimental CCS error, as it only uses the CCS as a filtering window. Our work proposes an improved alternative using the best parts of available tools for LC-IM-MS data and shows that using any of those alone would have not provided the same coverage and accuracy. To clarify this point, we have added the section “Comparing PeakDecoder to other workflows”. We expect that publishing this work will inform scientists on how to best process their LC-IM-MS data and will support our efforts to continue our research and to implement the algorithm as a new fully automated and enhanced tool.

9. Code Availability. Source code, demo code, example data, and detailed documents are required. It will help users to use this tool and reproduce results.

The source code and the data were all provided since the first version of the manuscript. Considering this suggestion, we have added further documentation in the GitHub page with details of the different input and output files and steps of the workflow.

10. Supplementary Figure S2: I'm curious about the linear range. Why the LC-IM-MS data have wider linear range than the SRM?

While it is possible that part of the improvement is due to the newer IM-MS instrument being more sensitive than the triple quadrupole instrument used, an important contribution to the wider linear range is due to the fact that with DIA in the IM-MS instrument all detectable fragments are recorded and all of them can be extracted post-acquisition and used for quantitation, therefore the sum of more fragments plus the intact precursor can increase the sensitivity compared to the sum of the fewer fragments used in the SRM method (typically 2-3). To clarify this point, we have added the following sentence to the Figure legend:

“The sum of more fragments plus the signal from the intact precursor in DIA can increase the sensitivity compared to the fewer fragments used in the SRM method (typically 2-3)”.

Minors:

11. Some typos:

Legend of figure 1. “...the query set of metabolites and estimate and FDR.” Remove second “and”.

The typo was corrected.

“Pseudomonas putida is a very promising industrial host,, with biochemical properties that make it ideal for hosting biochemical transformations”. Remove the replicated comma.

The typo was corrected.

Reviewer #2 (Remarks to the Author):

In “PeakDecoder: A Machine Learning-Based Metabolite Identification Algorithm for Multidimensional Mass Spectrometry Measurements and its Application in Synthetic Biology” the authors describe a machine learning-based feature detection and deconvolution tool for DIA LC-IM-MS data including FDR estimation and a nice demonstration of their tool for strain optimization experiments with different bacteria. While I think that the work and novel tools for metabolite annotation are very important and needed for metabolomics DIA and ion mobility data, there are a couple of important points I would suggest the authors should carefully consider and revise, perhaps with additional experiments, before the paper can be published.

While I am not an expert in machine learning, I have some doubts about the training data size and the generation thereof.

In particular, the use of another DIA deconvolution tool, but no ground truth, to generate the training data makes me wonder whether the tool is at risk of over-fitting and if it can perform better /be more sensitive than the initial tool.

Perhaps I have overlooked a critical discussion / comparison between the two. If not done yet, I would suggest doing this. Also a cross validation with other data sets, including ground truth (internal standards) that were not part of the training data might be useful.

While the training data size certainly affects the performance, PeakDecoder does not use the deconvolution results directly for training (please see the answer to point 2 of Reviewer #1). The discussion about the validity of our target-decoy generation method and comparison against existing ones is in the section “Developing the PeakDecoder algorithm”, paragraph:

Methods to generate decoys from experimental spectra have been previously reported, however, from a DDA MS/MS target library (i.e. annotated spectra), first in proteomics^{31,32} and more recently in metabolomics^{20,22}. We propose an alternative strategy to generate decoys taking advantage of the comprehensive nature of the DIA spectra. Instead of generating decoys from the target library, we perform UFD and TDX in the LC-IM-MS DIA data to generate a training set of peak-groups. The high-quality peak-groups constituted by the detected precursors (MS1) and its deconvoluted fragments (i.e., pseudo MS2) are used as targets. This strategy provides a noise-filtered ‘clean’ set of targets which was reported to be necessary to reach accurate estimates in spectrum level decoy-based methods²⁰. We then employ a pairing and swapping strategy, similar to the precursor-swap method proposed by Cheng et. al.³², but rather than swapping precursors, we generate the respective decoy precursors and fragments from the same targets by swapping pairs of fragment m/z (Fig. 1-c). Pairing precursors with the same number of fragments was used as an approach to increase the chances that the molecules are similar and to ensure that the overall distributions of general properties of targets and decoys are the same. Generating decoys by pooling and randomly adding fragments was avoided because it has previously shown poor performance (naïve method)²⁰, as it increases the probability of generating unrealistic decoys. Since the deconvoluted data represent real molecules, our decoy strategy is

valid in practice and the generated decoys comply with several conditions or properties, previously proposed for proteomics³¹, to calculate FDR with a valid target-decoy model: (i) the decoy library has the same precursor m/z and charge distributions as the target library, (ii) target and decoy spectra include the same number of peaks and have the same intensity sum distribution, and (iii) decoy spectrum peaks are positioned on realistic m/z values (fragments that naturally occur).

Importantly, none of the metabolites in the library are used for training (indicated in Fig. 1-b), as the library generated from standards (query set) is used only for inference in Step-6, where the model already trained is then used to assign metabolite annotation. We agree that an evaluation of the reliability of our FDR estimation method was very important and needed, and we thank Reviewer #2 for also pointing out this issue and suggesting internal standards. Besides the benchmarking against manually curated results as suggested by Reviewer #1, we have also included the results for an internal standard (tryptophan d5) in Supplementary Fig. 6-c.

I would also suggest discussing more critically the alternative use of DDA as a still widely used and established MS/MS strategy in metabolomics. Here it would have been nice to also compare the two. How many more metabolites can you ID by your DIA method and your in-house library in comparison to DDA and other public MS/MS libraries? Perhaps this could also serve to orthogonally confirm Metabolite IDs from PeakDecoder.

Several studies have reported comparisons of DDA vs. DIA both in proteomics and metabolomics, and we believe that such comparison is out of the scope of our work. To address this point, we have added a sentence and reference in the introduction:

Like initially found in proteomics⁹, in metabolomics the MS2 spectrum quality of ions that get selected during standard data-dependent acquisition (DDA) is higher, but the overall MS2 coverage and quantitative precision using DIA is better [Guo and Huan. 2020. 10.1021/acs.analchem.9b05135].

In addition, DDA would result in under sampling with our short 9 min LC method, and to indicate this we have added the following sentence in the section "Optimizing the LC-IM-MS analytical method":

DDA methods with short LC separation (<15 min) would be limited to only select the top 3-5 ions^{18,26} per cycle to preserve the MS1 sampling rate and quantitation dynamic range, which in turn would result in MS/MS under sampling of medium-low-abundance ions.

Regarding using public MS/MS libraries, the annotation confidence using MS/MS only will significantly decrease compared to our multidimensional approach which includes both RT and CCS. Ultimately, the number of IDs is not the most important metric to evaluate the usefulness of a method, and we believe that confident identifications are more important than the total number of IDs.

Furthermore, using public libraries (including experimental and predicted) could be performed but we are planning to do those as future work, since generating the

multidimensional library by predicting some of the dimensions to include all MS/MS, RT and CCS will require a significant amount of effort.

For the FDR estimation, this is great that the authors added this to their tool. Robust ways to estimate FDR in non-targeted metabolomics are certainly needed. While I like the simple concept of the precursor mass swapping method, I wonder how well this performs, also in comparison to other approaches.

Did you do any comparison here? Are there experience/infos available from benchmarking FDR metrics for PSM? I would suggest dedicating more attention to this part and critically discuss/evaluate this.

We were unable to perform a direct comparison to previously proposed methods for FDR assessment because they rely on large libraries of annotated MS/MS spectra. We have expanded the introduction including other FDR methods in metabolomics. In addition, we discussed the difference of our method compared to existing ones, performed the benchmarking against manually curated results shown in Fig. 2-c and included the results for an internal standard as Supplementary Fig. 6-c.

Detailed Comments:

Formatting:

I would suggest including line numbers for submissions of future/revised manuscripts. The line numbers are now included. We apologize that those were not included initially. The manuscript submission system automatically includes the line numbers for word files, but we had to submit the manuscript as a pdf due to rendering issues with the embedded figures and did not notice that the line numbers were missing.

Title:

Maybe better use “Metabolite Detection Algorithm” or “Metabolite Annotation Algorithm” as I assume your tool does not necessary provide Level 1 IDs for compounds detected. We agree that, in general, the term annotation should be used instead of identification. However, we would like to keep the word “identification” in the title, since our reported annotations do meet the data requirements for a confidence of “Level 1”, according to the Metabolomics Standards Initiative: at least two orthogonal techniques defining 2D structure confidently, such as MS/MS and RT or CCS (<https://doi.org/10.3390/metabo8020031>). Our library was built from pure standards analyzed by multidimensional separations and the annotations are based on MS/MS, RT and CCS (RT-CCS-DIA), or RT and CCS (RT-CCS).

Abstract:

You indicate that PeakDecoder™ is a Trademark. In the conflict of interest statement you further indicate that one of the Authors is an employee of Agilent. Perhaps it would be good to expand here and clarify how much Agilent is involved in software

development and IP and whether there could be any conflict of interest, particularly in bench-marking the tool.

To avoid confusion, we have removed the trademark symbol. There is no further conflict of interest to be declared since the algorithm is fully described and all software is open source and freely available. By definition, the use of the trademark symbol does not indicate registration and does not impart enhanced protections or financial interests. We were using it as a suggestion to the MS community that we would like the word "PeakDecoder" to be acknowledged and used only in association with our algorithm. Furthermore, the Author Contributions section specifies that the Agilent employee, Dr. Alex Apffel, contributed to the development and optimization of the LC method and the AMSS tool (not PeakDecoder), and Dr. Aivett Bilbao conceived the PeakDecoder algorithm and implemented the software.

Page 2 3rd paragraph:

According to your statement of "hundreds to thousands of primary metabolites and a multitude of secondary metabolites" one might assume that the chemical diversity is greater in primary metabolism, which I assume the authors did not want to claim. Perhaps, specify this more in detail also whether this is with regards to a single organism, or all metabolites in nature.

We agree that the sentence was confusing, and we have corrected it as suggested: "hundreds to thousands of primary and secondary metabolites in nature".

Page 3 paragraph 1:

I would suggest to point out more clearly the current limitation of DIA for metabolomics. Perhaps it would be good to mention studies that compare DIA to DDA.

Further, while I agree that there is no generally accepted method for FDR estimation in metabolomics, there have been several methods proposed. Perhaps it would be good to mention some of them and elaborate on the disadvantages/challenges here.

We have improved the Introduction and added additional references and comparisons, as also suggested by Reviewer #1.

Page 7 last paragraph:

What is the number of standards/ library size?

The library contains 64 standards, and this was already indicated in the Abstract, Results and Table 1. This is now also indicated in the Discussion.

Page 8:

In how many cases could you differentiate co-eluting metabolites. Did you compare the overall numbers of metabolite IDs to DDA experiments?

We did not perform a comparison to DDA. We have added the new Fig. 3 to compare the annotation selectivity by the different analytical separations in microbial samples. The number of possible LC-IM-MS peaks from MS-DIAL untargeted feature detection

results matched within tolerances was reduced the most when using all mass-RT-CCS dimensions. At the same time, using mass-RT or mass-CCS resulted in several matched features which are potentially other co-eluting and co-drifting metabolites differentiate by the multidimensional separation.

Page 9, 2nd paragraph.

“A total of 2350 metabolite features could be confidently annotated.” Do you mean annotated as metabolites or m/z – RT “Features”? If you mean features, perhaps “detected” would be the appropriate term here and should be changed throughout the manuscript.

If this refers truly to annotated metabolite IDs, this would be quiet amazing. I would suggest to provide the table of those compounds inc. score, mass error etc. in the SI akin to SI Table 1, where so far only 66 metabolites are listed.

We agree that this could be misleading. We have added the word detected: “features could be detected and confidently annotated.” In addition, we are using the term “feature” instead of “metabolite feature” throughout the manuscript to avoid confusion. All the results including the annotations were already provided in the GitHub repository. As suggested, we have now added Supplementary Tables 2-4, to show unique metabolites identified and scores from the best sample.

Figure 1 and Page 9, 2nd paragraph:

More details need to be given here. How were the “best quality detected” defined, what is the typical size of training data and perhaps visualize in the figure how the decoys are generated

The best quality is defined by the XIC metrics that are used to filter high-quality fragments and to keep only high-quality targets, as specified in Methods “Machine learning training”: precursor S/N > 20, and at least 2 fragments with mass error < 15 ppm, RT difference to their precursor < 0.1 min, and FWHM difference to their precursor < than 2x the precursor FWHM.

To clarify this point, we replaced the sentence “best quality detected” by:

Before training, filtering for high-quality fragments is applied to keep high-quality peak-groups as targets (i.e., based on various thresholds for metrics of precursor and at least 3 fragments: S/N, mass error, RT difference to precursor, and FWHM difference to precursor; details in Methods) and their corresponding decoys in the final training set.

Regarding the typical size, the new Supplementary Fig. 5 shows that we could get accuracy > 98.86 if the resulting training set contained between 2760 and 6720 targets. As suggested, we have placed the example of decoy generation as Fig. 1-c.

Table 1:

See my comment above regarding the term “annotated”. Also did you compare the annotation / Identification rate to more standard DDA approaches? From my own experience I would assume that typically more than 28 metabolites should be IDed

when using high-quality data and contemporary spectral libraries (Massbank, NIST2020, Metlin, GNPS etc.).

Did you match your deconvoluted DIA spectra against any public MS/MS libraries, or only against the in house library. Perhaps other DIA databases such as “Weizmass” (Shahaf et al. Nat Com) might also be relevant here.

We are now using the terms “# unique metabolites” and “# annotated features” in Table 1. We appreciate the suggestion and agree that the number of IDs will increase using larger public libraries. But as mentioned previously, the annotation confidence using only MS/MS will significantly decrease compared to our multidimensional approach which include RT and CCS. As pointed out by Reviewer # 3, rather than the number of IDs, accurate identification of metabolites is important for many different application scenarios.

Page 13:

How was the quant experiments for metabolites / proteins performed. Internal /external standards?

Do you have absolute concentrations? (Like for Bisabolene) Or do you refer to relative quant? Perhaps state this in the text / figure title and caption.

All the metabolomics results reported are relative levels and label free except for the bisabolene production measurement in a dodecane overlay which was done as absolute concentration (units of mg/L provided, Fig 5.b). We have now specified this in all the figures. We also noted that this panel of extracellular bisabolene was repeated, and we have fixed this error.

Figure 4:

The quantification refers to relative quant or absolute quant with authentic standards.

The quantification is relative and label free. We have now specified this in all the figures.

Also why is the here presented set of compounds smaller than the list in the SI or the above mentioned 2350 metabolite features.

The features count repetitions, i.e., the same metabolite identified in multiple samples.

The total number of features are counted across all 3 different datasets.

Page 16, Discussion:

Why will LC-IM-MS and PeakEncoder enable faster testing of strains? I would argue that conventional DDA LC-MS/MS and other state of the art feature finding tools will result in the same speed. On the other hand, MS1 based approaches such a FIA or Rapidfire are likely significant faster. Before overstating speed here, the advantages should eb clearly discussed and perhaps typical analysis/processing times should be given.

We agree that this statement needs correction and elaboration, and we thank Reviewer #2 for the suggestion. LC-IM-MS and PeakDecoder enable faster testing of strains with increased metabolite coverage and more confident identifications.

While it is true that the same LC method with other MS instrumentation will result in the same speed, our analytical and computational workflow provides faster and better results due to several aspects which we now enumerate in the discussion as follows:

The advantages of using LC-IM-MS with DIA and PeakDecoder enable high-throughput analyses with increased metabolite coverage and more confident identifications due to several aspects: 1) our 9 min LC method is faster than the GC methods typically used, 2) the ion mobility dimension further separates more analytes and increases annotation confidence by combining CCS and RT compared to LC alone, 3) DIA further increases annotation confidence with fragmentation information and provides better reproducibility and dynamic range than DDA, and 4) Our PeakDecoder score provides a confident metric for metabolite annotation.

We have also added the following sentence and reference in the Results:

Compared to the methods typically used to perform GC-MS-based global metabolomics²⁴, this LC method provides a ~3x faster sample analysis time.

Page 17 2nd paragraph:

Awesome that you made the library available to the community. Perhaps infos about file format and a download link would be nice to have here too.

We have initially provided the library as csv file in the GitHub repository and now added them as msp files, and these files are also included as a zip file as supplementary information. We have added a sentence in the Results:

“The list of metabolites can be found in Supplementary Table 1 and the library can be found as csv and as msp format as Supplementary-File1.zip.”

Reviewer #3 (Remarks to the Author):

The authors propose a combination of analytical and computational methods for metabolite profiling based on LC-IMS-MS in DIA mode. A main contribution is the so-called PeakDecoder-algorithm to score library hits. The algorithm consists of several stages, some of them based on support vector machines.

Particularly interesting is the strategy to generate decoys, which seems logical and probably generalizable to other fields apart from metabolomics that should be just as appropriate as the strategy layed out in reference 20 of the proposed manuscript.

Considering how important accurate identification of metabolites is for many different application scenarios, I would expect the work to have significant impact in the field.

The methodology is sound and the conclusions are supported by the analysis and the

presented data. The computational building blocks are comparably simple. While it makes sense, in my opinion, to err on the side of simplicity rather than complexity, a sensitivity analysis for some of the choices (mostly cosine distance for comparison and SVMs as classifier) would have been nice, but I strongly suspect that it would not change that much.

In summary, I consider the proposed workflow as a very interesting alternative to the state of the art with the potential of significant impact, and would recommend to accept the manuscript for publication.

We are very grateful for the comments of Reviewer #3, and we are thrilled because the reviewer noticed and appreciated the potential for generalization of our algorithm and applicability to other molecular types.

We agree that a sensitivity analysis could be useful, but rather than optimizing traditional machine learning models, we would like to invest our efforts in implementing a fully automated workflow based on deep learning.

REVIEWER COMMENTS

Reviewer #1 (Remarks to the Author):

In the revised manuscript, the authors have fully addressed my concerns. I really appreciate the authors' continuous efforts to develop IM-MS tools. Congratulations!

Reviewer #2 (Remarks to the Author):

Thanks a lot for responding to my comments and for the revision of your manuscript. Most of my comments have been well addressed and/or clarified. However, a couple of points are still remaining / require further clarification and I would suggest the authors to incorporate them before the manuscript gets published.

Unfortunately, I did not find a revised manuscript file with highlighted/tracked changes, which made it sometimes hard to see what you have changed, and I might have overlooked some changes the authors did.

Response 1:

I am not sure why the authors quoted the method paragraph about FDR here, as my concern was about the training data for their ML model. This should be more critically discussed and potential limitations should be clearly pointed out. E.g. what was your training data size and what would you recommend as minimum requirements to avoid over-fitting?

While I appreciate the addition of one internal standard and manual curated results for validation, maybe a more systematic/large scale approach would add more confidence. Did you consider using a larger set of standards or using DDA spectra for validation?

Response 2:

I agree with the authors that many studies have shown the comparison of DDA vs. DIA. However unlike in proteomics, the advantages in metabolomics are not that clear. Important difference in metabolomics are for example the limited spectral library size, limited confidence in silico fragmentation methods and, in general, the more diverse fragmentation behavior of the larger chemical space. Hence, I don't think the advantage of DIA in metabolomics is as clear as the authors present it, and these points should be critically pointed out.

As the authors are probably aware, a common strategy in DIA-based proteomics is the use of in-house spectral libraries for the biological system of interest. Such libraries are often generated with iterative / pre-fractionated DDA runs, which provides the highest level of annotation confidence.

That being said, I am convinced that a DDA-based spectral library and its use as ground truth for your deconvolution would be highly beneficial for the validation of your approach.

As stated in my original comment, I think it would be very interesting to see what the differences in terms of compound annotation and feature coverage between the two (DDA vs. DIA+PeakDecoder) approaches are, and your paper would benefit from such a comparison.

Response 3:

Thanks for expanding the introduction. Unfortunately, I could not find a highlighted version of your revised manuscript. Please provide specific line numbers our quote the text in your response letter.

As for the benchmarking with manual curated results, what was the number of those, and do you think this and one internal standard is representative?

Minor Comment:

For the title, I think that "annotation" would still be more appropriate as most "features" you detect are probably not level 1 IDs, or does PeakDecoder only output those with CCS/RT/ms2 matches?

Regarding the discussion about speed, I don't think it makes sense to compare your approach to GC-MS runs/processing. I would suggest to either remove this statement or compare the processing speed of PeakDecoder to other DIA data processing methods.

Response to reviewers

Manuscript ID: NCOMMS-22-26518A

Title: "PeakDecoder enables machine learning-based metabolite annotation and accurate profiling in multidimensional mass spectrometry measurements"

Authors: Aivett Bilbao, Nathalie Munoz, Joonhoon Kim, Daniel J Orton, Yuqian Gao, Kunal Poorey, Kyle R. Pomraning, Karl Weitz, Meagan Burnet, Carrie D. Nicora, Rosemarie Wilton, Shuang Deng, Ziyu Dai, Ethan Oksen, Deepti Tanjore, James Gardner, Richard D. Smith, Joshua K. Michener, John M. Gladden, Erin S. Baker, Christopher J. Petzold, Young-Mo Kim, Alex Apffel, Jon K. Magnuson and Kristin E. Burnum-Johnson

Authors' response: We are grateful to the reviewers for taking the time again to review our revised manuscript. We have further revised our manuscript and addressed the additional points as described in the following sections (authors' response in blue text).

Note: following the suggestion of Reviewer #2, the word "identification" was replaced by "annotation" in the updated title "PeakDecoder enables machine learning-based metabolite annotation and accurate profiling in multidimensional mass spectrometry measurements".

Reviewer #1 (Remarks to the Author):

In the revised manuscript, the authors have fully addressed my concerns. I really appreciate the authors' continuous efforts to develop IM-MS tools. Congratulations!

Thank you very much! Again, we greatly appreciate your time and feedback which allowed us to substantially improve the presentation and robustness of our PeakDecoder workflow.

Reviewer #2 (Remarks to the Author):

Thanks a lot for responding to my comments and for the revision of your manuscript. Most of my comments have been well addressed and/or clarified. However, a couple of points are still remaining / require further clarification and I would suggest the authors to incorporate them before the manuscript gets published.

Unfortunately, I did not find a revised manuscript file with highlighted/tracked changes, which made it sometimes hard to see what you have changed, and I might have overlooked some changes the authors did.

We thank you for the additional suggestions and apologize for the issue finding the revised document with the tracked changes. We understand that this is very important to facilitate the reviewing process. We did upload a Word version as file type “Related manuscript file” where all changes were tracked (using the Word functionality to track changes) but did not notice that the submission system converted it to PDF ignoring the tracked changes. We have inquired to the editor to make sure that this second revision properly includes the change-tracked file for review. Reviewers should be able to find the text version with tracked changes in a PDF file containing the word “_related_ms_” in the file name.

Response 1:

I am not sure why the authors quoted the method paragraph about FDR here, as my concern was about the training data for their ML model. This should be more critically discussed and potential limitations should be clearly pointed out. E.g. what was your training data size and what would you recommend as minimum requirements to avoid over-fitting?

We quoted the FDR method because it is related to how the training data is generated. Discussions regarding the training data size, overfitting and limitations, were also already addressed in the first revision, but we recognize that those were not specifically included in the original response for this point of Reviewer #2.

Please see the paragraph at page 8, line 9, specifying how the size and quality of the training data can be controlled (which depends on the untargeted feature detection and deconvolution tool used, referred to as UFD):

... the performance of PeakDecoder depends on the training set and the validity of the estimated FDR depends on the number of generated false positives. The size and quality of the training set can be controlled in two ways: the parameters of the UFD tool used to generate the preliminary training set (**Fig. 1-b**, Step-1) and the filtering for high-quality fragments used to generate the final training set (**Fig. 1-b**, Step-4). At the same time, a tradeoff in the quality of peak-groups is necessary to avoid overfitting and perfect training accuracy, and thus, to estimate a reliable FDR. These components allow the user to define the quality of the resulting annotations and are evaluated using microbial data in the next section.

See the paragraph at page 10, line 34, specifying that, from our results, training data contained between 2760 and 6720 targets was sufficient for good training performance:

Supplementary Fig. 5 shows that the training performance was not significantly impacted by the deconvolution parameters if the numbers of targets was sufficient (accuracy > 98.86 if the resulting training set contained between 2760 and 6720 targets), but at the same time, if the classifier resulted in a close-to-perfect accuracy (>99), the minimum non-zero FDR that could be estimated was affected due to the small number of false positives.

And see the paragraph at page 21, line 33, in the Discussion, regarding limitations: Limitations of the current algorithm include requirements for sufficient high-quality peak-groups for training (i.e., limited performance for samples with very low complexity) and a library acquired with compatible analytical conditions for inference.

While I appreciate the addition of one internal standard and manual curated results for validation, maybe a more systematic/large scale approach would add more confidence. Did you consider using a larger set of standards or using DDA spectra for validation?

We agree that a more systematic and large-scale comparison including DDA would be very useful, but due to several reasons related to project scope and sample limitations, we performed the comparison against the library built from authentic standards.

Response 2:

I agree with the authors that many studies have shown the comparison of DDA vs. DIA. However unlike in proteomics, the advantages in metabolomics are not that clear. Important difference in metabolomics are for example the limited spectral library size, limited confidence in silico fragmentation methods and, in general, the more diverse fragmentation behavior of the larger chemical space. Hence, I don't think the advantage of DIA in metabolomics is as clear as the authors present it, and these points should be critically pointed out.

We agree that not all the advantages of DIA in metabolomics are directly translated from DIA in proteomics, particularly in terms of data processing tools. The advantages cited in the Introduction specifically refer to data quality: MS2 coverage and quantitative precision (page 3, line 5). Precisely because there are other limitations in DIA in metabolomics, such as spectral libraries and in-silico fragmentation, we developed PeakDecoder, which proposes a different data processing strategy and multidimensional measurements including CCS. Please see the paragraph at page 12, line 31:

Our decoy strategy for DIA data together with IM and LC conveys a powerful multidimensional characterization of metabolites that address several important challenges. For many metabolites only a few characteristic fragment ions can be detected, rendering the use of classic spectral similarity searches unreliable¹⁸. Moreover, some metabolites could not be detected with even a single fragment. In these cases, the CCS increased the identification confidence compared to using the RT and accurate mass alone.

As the authors are probably aware, a common strategy in DIA-based proteomics is the use of in-house spectral libraries for the biological system of interest. Such

libraries are often generated with iterative / pre-fractionated DDA runs, which provides the highest level of annotation confidence.

That being said, I am convinced that a DDA-based spectral library and its use as ground truth for your deconvolution would be highly beneficial for the validation of your approach.

As stated in my original comment, I think it would be very interesting to see what the differences in terms of compound annotation and feature coverage between the two (DDA vs. DIA+PeakDecoder) approaches are, and your paper would benefit from such a comparison.

We agree that a comparison including DDA would be very interesting and useful, but unfortunately, we do not have samples left to perform additional DDA analyses in this work. To clarify this point, we added the text in red in the following sentence (page 21, line 43) in the Discussion:

Future work will be performed to compare PeakDecoder to DDA analyses and to evaluate it with predicted MS/MS, CCS and RT metabolomics libraries.

Response 3:

Thanks for expanding the introduction. Unfortunately, I could not find a highlighted version of your revised manuscript. Please provide specific line numbers our quote the text in your response letter.

The paragraphs added in the Introduction (page 3, line 7 and 24) are the following ones:

Two main DIA processing strategies initially established for proteomics have been adapted to metabolomics in a handful of DIA metabolomics tools. The first strategy applies untargeted feature detection followed by deconvolution of fragment ion spectra (here referred to as UFD). A popular tool used for UFD in metabolomics is MS-DIAL¹¹, which groups precursors and their corresponding fragments based on the similarity of their elution profiles, generates pseudo-MS2 spectra and matches them against a reference MS2 library. Other reported tools applying UFD are MetaboDIA¹² and DaDIA¹³. The second DIA algorithmic strategy employs targeted data extraction (here referred to as TDX). TDX requires a library of target analytes with retention times, and precursors with corresponding fragment masses, which are utilized as coordinates to mine the DIA spectra and generate extracted ion chromatograms (XIC) for precursor and fragments per target analyte, as the so-called 'peak-group'. Multiple sub-scores are then calculated per peak-group to assess coelution and identification. Software employing TDX include Skyline¹⁴, MetDIA¹⁵ and DIAMetAlyzer¹⁶. Another tool demonstrated for DIA using a different approach is DecoID¹⁷, where the MS2 deconvolution is achieved by mixing database spectra to match an experimentally acquired spectrum using least absolute shrinkage and selection operator (LASSO) regression.

While these tools exist for DIA metabolomics, new tools capable to fully exploit all dimensions with controlled error rates in multidimensional LC-IM-MS measurements with DIA spectra are needed. Skyline and MS-DIAL were adapted to support the additional IM dimension but they do not provide a false-discovery rate (FDR) control method. Unlike proteomics, the field of metabolomics still lacks a generally accepted, validated, and automated calculation of error rates for MS2 compound identification with FDR assessments¹⁸. Several methods have been proposed to generate decoys and estimate FDR in metabolomics. For imaging-MS, pySM¹⁹ generates decoys by using implausible ion adducts. For DDA, Passatutto²⁰ uses re-rooted fragmentation trees, JUMPm²¹ adds a small odd numbers of hydrogen atoms, and XY-Meta²² combines original and randomly selected MS2 peaks. And recently reported for DIA, DIAMetAlyzer¹⁶, provides an FDR estimation employing Passatutto²⁰ but it does not support the IM separation. These methods rely on annotated spectra or a sample-specific metabolite database for FDR estimation.

As for the benchmarking with manual curated results, what was the number of those, and do you think this and one internal standard is representative?

We thank the Reviewer # 2 for pointing this out, we added the text in red in the following sentence (page 12, line 22) in the section “Comparing PeakDecoder to other workflows”:

we performed a comparison against the ground truth generated from manually curating the full *P. putida* dataset, with 550 peak-groups including 233 positives and 317 negatives.

And we agree that one internal standard alone is not sufficient to properly validate an approach, but it serves as an example and supplements the rest of the evaluations that we performed.

Minor Comment:

For the title, I think that “annotation” would still be more appropriate as most “features” you detect are probably not level 1 IDs, or does PeakDecoder only output those with CCS/RT/ms2 matches?

The word “identification” was replaced by “annotation” in the title. As indicated in the manuscript (e.g., legend of Table 1), PeakDecoder outputs both types of matches by accurate mass plus: either 2 dimensions with RT-CSS or 3 dimensions with RT-CCS-DIA (MS2).

Regarding the discussion about speed, I don’t think it makes sense to compare

your approach to GC-MS runs/processing. I would suggest to either remove this statement or compare the processing speed of PeakDecoder to other DIA data processing methods.

The comparison to GC-MS refers to data acquisition, not to data processing. To clarify this point, we added the text in red in the following sentence (page 21, line 4) in the Discussion:

1) **in terms of data acquisition**, our 9 min LC method is faster than the GC methods typically used.